# Smith-specific regulatory T cells halt the progression of lupus nephritis

Peter J. Eggenhuizen [1,8], Rachel M. Y. Cheong [1,8], Cecilia Lo[1], Janet Chang[1], Boaz H. Ng[1], Yi Tian Ting[1], Julie A. Monk[1], Khai L. Loh [1], Ashraf Broury [1], Elean S. V. Tay[1], Chanjuan Shen[2], Yong Zhong[1,3], Steven Lim [4], Jia Xi Chung[1], Rangi Kandane-Rathnayake[1], Rachel Koelmeyer[1], Alberta Hoi[1,5], Ashutosh Chaudhry[6], Paolo Manzanillo[7], Sarah L. Snelgrove [1], Eric F. Morand [1,5] & Joshua D. Ooi [1] ✉

Antigen-specific regulatory T cells (Tregs) suppress pathogenic autoreactivity and are potential therapeutic candidates for autoimmune diseases such as systemic lupus erythematosus (SLE). Lupus nephritis is associated with auto-reactivity to the Smith (Sm) autoantigen and the human leucocyte antigen (HLA)-DR15 haplotype; hence, we investigated the potential of Sm-specific Tregs (Sm-Tregs) to suppress disease. Here we identify a HLA-DR15 restricted immunodominant Sm T cell epitope using biophysical affinity binding assays, then identify high-affinity Sm-specific T cell receptors (TCRs) using high-throughput single-cell sequencing. Using lentiviral vectors, we transduce our lead Sm-specific TCR into Tregs derived from patients with SLE who are anti-Sm and HLA-DR15 positive. Compared with polyclonal mock-transduced Tregs, Sm-Tregs potently suppress Sm-specific pro-inflammatory responses in vitro and suppress disease progression in a humanized mouse model of lupus nephritis. These results show that Sm-Tregs are a promising therapy for SLE.

Regulatory T cells (Tregs) are essential for maintaining immune homeostasis by acting in a variety of ways on different immune cell types (Fig. 1). Decreased Treg numbers and/or defective Treg function have been implicated in the development of autoimmune diseases[1], such as systemic lupus erythematosus (SLE)[2,3]. Treg-based therapeutic approaches, such as expansion of polyclonal[4] or antigen-specific Tregs[5], are increasingly being explored, and antigen-specific Tregs, in particular, have been shown to suppress pathogenic autoreactivity[6].

Lupus nephritis (LN), a severe manifestation of SLE, is an important contributor to disease related morbidity and mortality[7]. The presence of LN is strongly associated with autoreactivity to the Smith

(Sm) autoantigen[8], which is in turn strongly associated with carriage of the human leukocyte antigen (HLA) haplotypes DRB1*15:01 (DR15) and DRB1*03:01 (DR3)[9–11]. Whether these associations are causal, and afford a therapeutic opportunity, can be explored through the design of an Sm antigen-specific Treg. Here, we develop a platform to establish autoantigen-specific Treg-cell-based therapy for autoimmune diseases (Fig. 2). In LN, we identify Sm HLA-DR15 restricted CD4[+] T-cell epitopes and highly reactive TCRs of the most immunogenic epitopes, create antigen-specific Tregs for the Sm autoantigen (Sm-Tregs), and evaluate the capacity of Sm-Tregs to suppress disease activity in vitro and in a humanized mouse model of lupus nephritis.

[1]Centre for Inflammatory Diseases, Department of Medicine, School of Clinical Sciences, Monash University, Clayton, VIC, Australia. [2]Department of Hematology, The Affiliated Zhuzhou Hospital of Xiangya Medical College, Central South University, Zhuzhou, China. [3]Department of Nephrology, Xiangya Hospital, Central South University, Changsha, China. [4]Alfred Research Alliance Flow Cytometry Core Facility, Melbourne, VIC, Australia. [5]Department of Rheumatology, Monash Health, Clayton, VIC, Australia. [6]Former Employee of Amgen, South San Francisco, CA, USA. [7]Amgen Research, Amgen Inc, South San Francisco, CA, USA. [8]These authors contributed equally: Peter J. Eggenhuizen, Rachel M. Y. Cheong. ✉e-mail: Joshua.Ooi@monash.edu

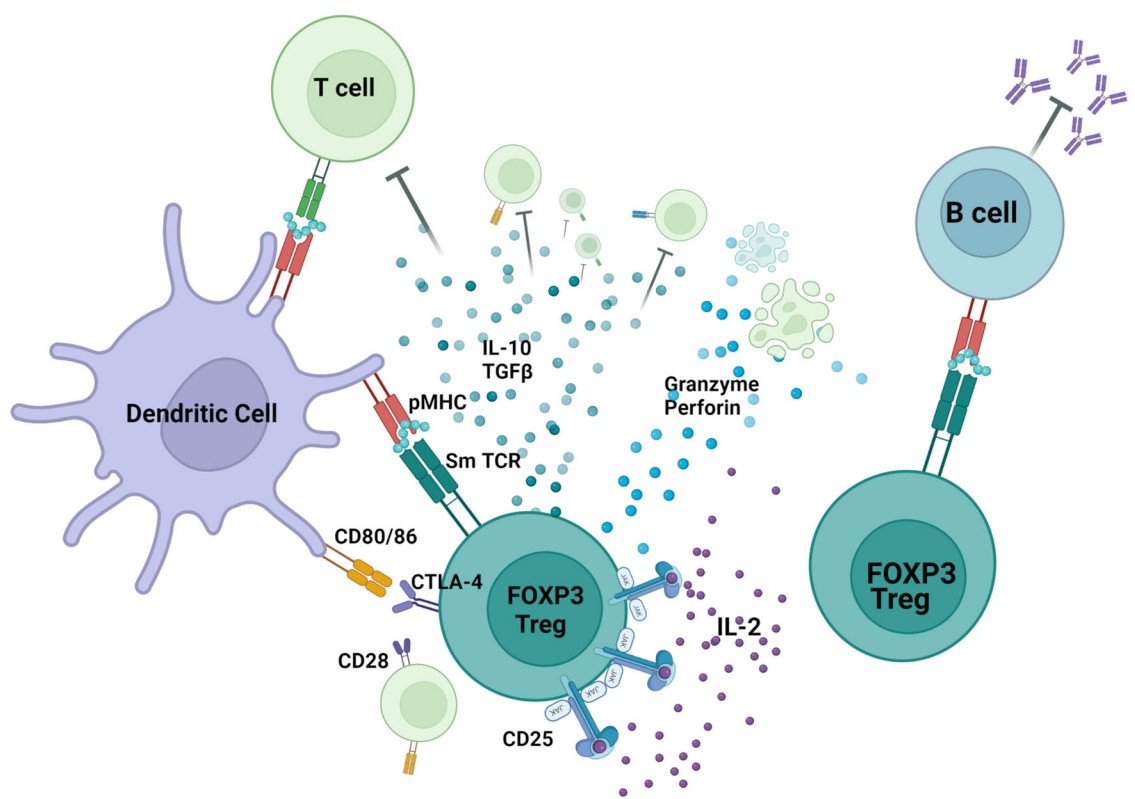

**Fig. 1 | Sm-TCR Treg mechanisms of action.** Sm-TCR Tregs act through a variety of mechanisms to suppress autoreactivity and restore immune tolerance. Tregs can suppress autoreactive B cells and autoantibody production, suppress pathogenic T cells, tolerize antigen-presenting cells such as dendritic cells, and can directly lyse inflammatory cells. TCR transduction of a Sm-specific TCR renders the Treg more potent in its ability to specifically suppress lupus nephritis. Created with BioRender.com.

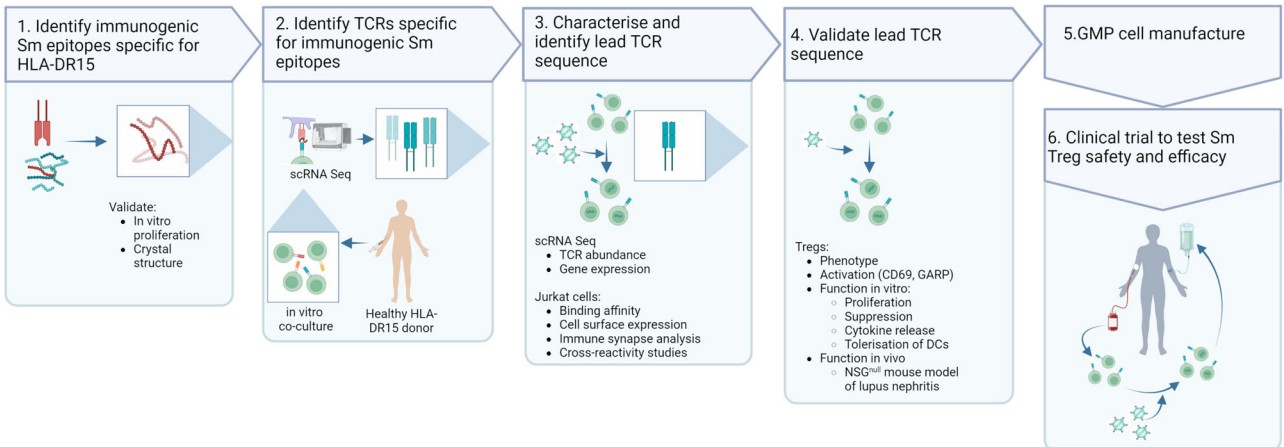

**Fig. 2 | Pipeline for the development of antigen-specific Tregs for autoimmune disease.** To develop an antigen-specific Treg targeting an autoimmune disease a 6-step process is undertaken. In the case of lupus nephritis, Sm-Tregs are developed as follows. (1) The immunogenic autoimmune epitope, the Smith antigen (Sm), in the context of human leukocyte antigen (HLA)-DR15 is identified. (2) T-cell receptors (TCRs) are identified specific for Sm. (3) Candidate TCRs are screened for high expression, specificity and reactivity and a lead Sm TCR is chosen. (4) Lead Sm TCR is validated for its functional responses on Tregs in vitro and in vivo. (5) The pre-clinical data package is sent for regulatory approval and good manufacturing practice (GMP) cell manufacture commences. (6) Sm-Tregs are tested in human clinical trials for safety and efficacy. Created with BioRender.com.

## Results

### Immunogenic Smith epitope discovery

We used a biophysical affinity binding assay to screen 145 overlapping 15-mer peptides spanning the key Sm proteins: SmB/B', SmD1, and SmD3[12] (Fig. 3a–c). The top five Sm epitopes with strongest binding to HLA-DR15 were: SmB/B'$_{1-15}$: KMLQHIDYRMRCILQ; SmB/B'$_{58-72}$: RVLGLVLLRGENLVS; SmD3$_{43-57}$: MSNITVTYRDGRVAQ; SmD3$_{61-75}$: VYIRGSKIRFLILPD; SmD1$_{46-60}$: TLKNREPVQLETLSI (Fig. 3d). Of all the Sm peptides tested, SmB/B'$_{58-72}$ had the highest binding to HLA-DR15 as measured by the % binding relative to a unique positive control peptide bound to HLA-DR15. Of all the Sm peptides tested, SmB/B'$_{58-72}$ also showed the greatest stability for HLA-DR15 as measured by the Stability Index, which measures the stability of each peptide-HLA−DR15 complex as a value normalized to the half-life of the

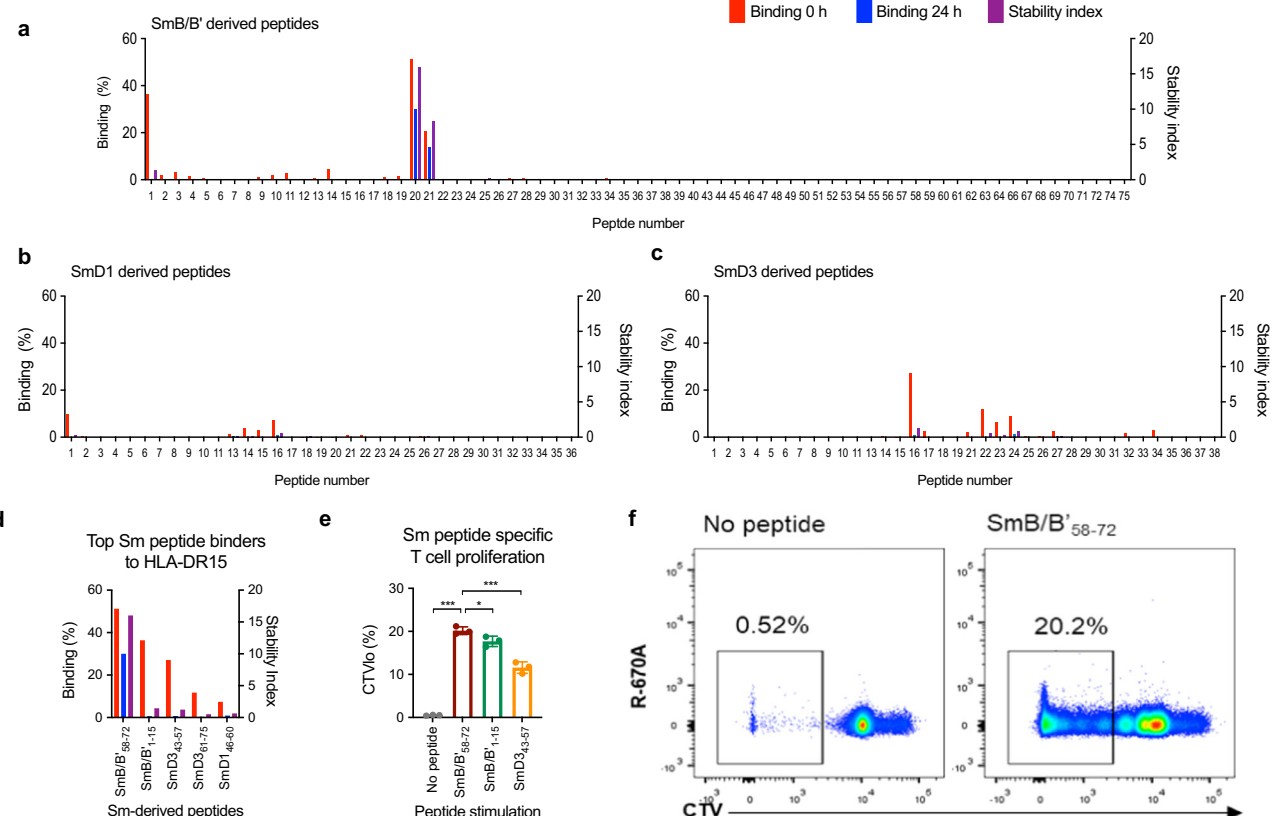

**Fig. 3 | Identification of SmB/B'₅₈₋₇₂ as the dominant T-cell epitope in lupus nephritis.** Physical binding affinity assay of HLA-DR15 and 145 synthesized 15-mer peptides derived from **a** SmB/B', **b** SmD1, and **c** SmD3. X axis shows peptide numbering of the 15-mers sequentially overlapping by an offset of three amino acids starting from the N-terminus of each protein. Y axes showing % binding and stability index, as described in "Methods". **d** The top five binding HLA-DR15-restricted Sm peptides from the physical binding assay. **e** Immunogenicity of SmB/

B'₅₈₋₇₂, SmB/B'₁₋₁₅ and SmB/B'₄₃₋₅₇ measured by the proportion of CD4⁺ CellTrace Violet (CTV)^lo T cells following 6-day co-culture of dendritic cells and CTV-labeled CD4⁺ T cells with or without peptide (n = 3 independent samples, data are presented as mean with SD) **f**, representative FACS plots for immunogenicity of SmB/B'₅₈₋₇₂-stimulated and un-stimulated CD4⁺ T cells by CTV dilution. Source data are provided as a Source Data file.

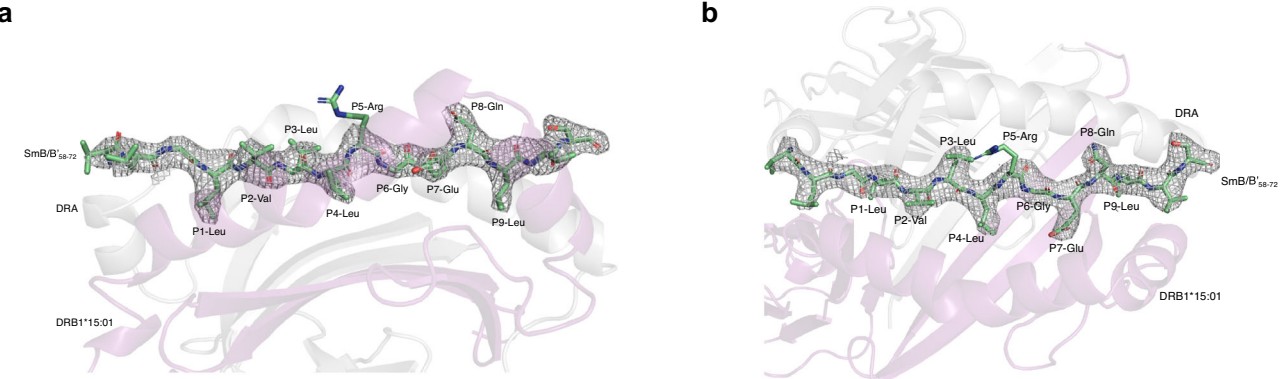

**Fig. 4 | Crystal structure of SmB/B'₅₈₋₇₂ bound to HLA-DR15. a** TCR top view of the DR15 bound SmB/B'₅₈₋₇₂ with 2 *Fo-Fc* electron density map, contoured at 1σ. **b** The peptide is bound in an extended conformation across the DR15 peptide binding groove (side view) with P2-Val, P3-Leu, P5-Arg, and P8-Gln accessible to TCR.

peptide-HLA and the binding score (see "Methods"). The top three Sm peptides were tested for immunogenicity by in vitro proliferation assay and SmB/B'₅₈₋₇₂ induced the strongest T-cell proliferation response (Fig. 3e, f).

**Crystal structure of Smith epitope-HLA-DR15 complex**
To visualize the binding of the SmB/B'₅₈₋₇₂ epitope to HLA-DR15 and to identify the amino acid residues that drive a T-cell response, we solved

the protein crystal structure of SmB/B'₅₈₋₇₂ in complex with HLA-DR15. The structure was solved at 3.12 Å, R_wor ~19.79% and R_free ~ 25.74%. The 9mer peptide binding core was LVLLRGENL, with P2-Val, P3-Leu, P5-Arg, and P8-Gln accessible to TCR binding (Fig. 4). Similar to previously solved HLA-DR15 protein structures in complex with myelin basic protein (MBP₈₅₋₉₉) in multiple sclerosis[13] and the alpha 3 subunit of type IV collagen (α3₁₃₅₋₁₄₅) in Goodpasture's disease[6], SmB/B'₅₈₋₇₂ bound favorably to the P1 and P4 hydrophobic peptide binding pocket.

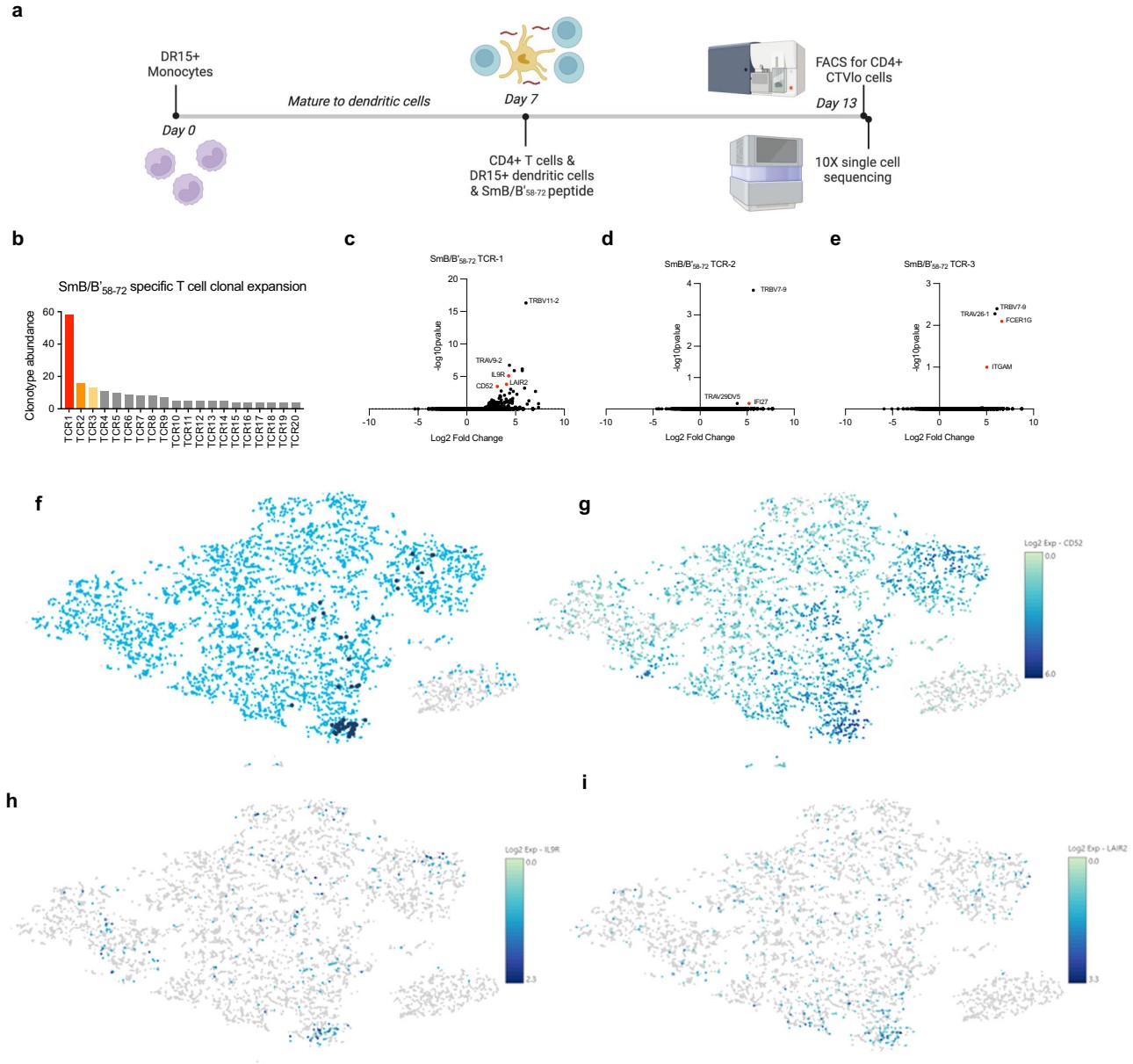

**Fig. 5 | Single-cell RNA sequencing reveals SmB/B′$_{58\text{-}72}$-specific CD4$^+$ T cells.**
**a** Experimental timeline of single-cell sequencing experiment from co-culture of cells with the peptide of interest to sorting of cells for sequencing. Created with BioRender.com. **b** Clonotype numbers of the top 20 SmB/B′58-72-specific TCRs (TCR1 to TCR20) as identified using 10X V(D)J single T-cell sequencing. **c–e** Volcano plot of genes expressed by CD4$^+$ T cells of interest that express SmB/B′$_{58\text{-}72}$-TCR1-3. *P* values are derived from a negative binomial exact test with adjustment using Benjamin Hochberg correction for multiple tests. **f** t-SNE plot of CD4$^+$ T cells that express our TCR of interest, TCR1 (dark blue dots). The majority of the cells expressing this TCR are clustered close together, indicating they are clonally expanded sharing a similar gene expression profile. **g** t-SNE plot of CD52 expression, a marker of suppressor T cells. The majority of CD52$^{hi}$ cells are clustered towards the bottom of the plot where cells expressing our TCR1 of interest are. **h** t-SNE plot of IL9R expression clustered with TCR1 expression. **i** t-SNE plot of LAIR2 expression clustered with TCR1 expression. Source data are provided as a Source Data file.

## Identification of Smith-specific T-cell receptors

To identify high-affinity SmB/B′$_{58\text{-}72}$-specific TCRs, we combined the use of high-throughput antigen-specific TCR single-cell sequencing, dextramer binding affinity assays, and imaging flow cytometry. First, we obtained peripheral blood mononuclear cells (PBMCs) from an HLA-DR15 homozygous healthy donor and performed high-throughput single-cell TCR sequencing on activated CD4$^+$CTV$^{lo}$ T cells co-cultured with monocyte-derived dendritic cells and the SmB/B′$_{58\text{-}72}$ peptide (Fig. 5a). The most abundant SmB/B′$_{58\text{-}72}$-specific TCR clonotype, TCR1, had 59 clones, >3 times more abundant than the next two clonotypes, TCR2: 16 clones, TCR3: 13 clones (Fig. 5b and Supplementary Table 1).

These TCRs did not show TCR clonal expansion when the same donor cells were co-cultured with a different 15-mer peptide (SmB/B′$_{1\text{-}15}$). The apparent relative affinities of the top three SmB/B′-specific TCRs were compared using a flow cytometry-based dextramer binding assay. TCR1 had the strongest affinity for SmB/B′$_{58\text{-}72}$ ($K_D$ = 0.3910 nM; B$_{max}$ of 3157), followed by TCR2 ($K_D$ = 0.9908 nM), then TCR3 ($K_D$ = 1.235) and negative control dextramer of DR15/ CLIP$_{103\text{-}117}$ showed very weak affinity ($K_D$ = 72.60) (Fig. 6b). Significantly upregulated genes associated with T-cell activation were *CD52, IL9R,* and *LAIR2* in CD4$^+$ T cells expressing TCR1, *IFI27* in cells expressing TCR2, and *FCER1G* and *ITGAM* in cells expressing TCR3 (Fig. 5c–i).

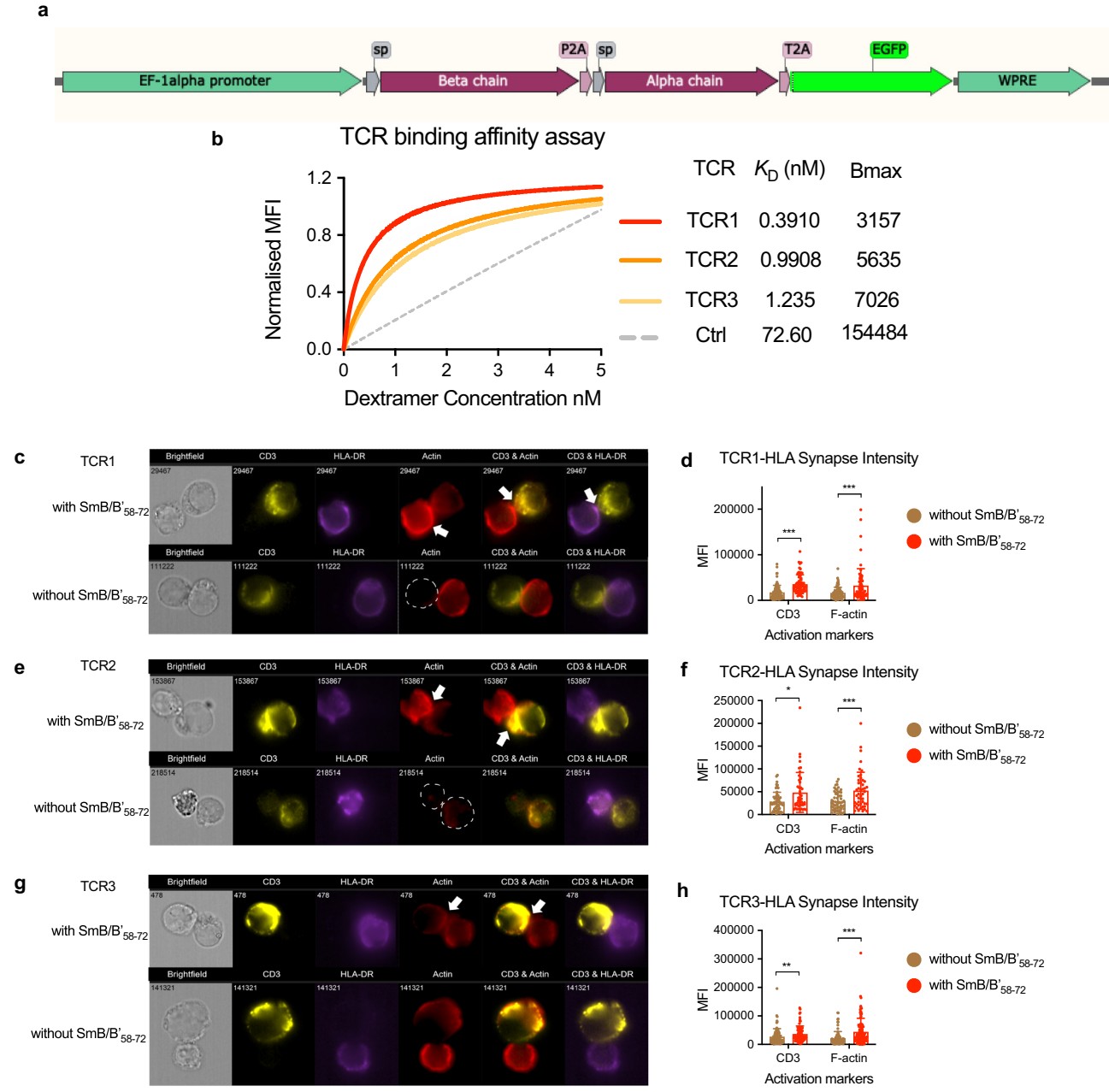

**Fig. 6 | Sm-TCR lentiviral transduction produces high-affinity, antigen-specific responses. a** lentiviral construct design showing from 5′ to 3′ the Sm-TCR beta chain, P2A ribosomal skipping sequence, Sm-TCR alpha chain, T2A ribosomal skipping sequence, and eGFP reporter under the control of a 5′ EF1-alpha promoter and 3′ WPRE. **b** Saturation binding curves of the top three SmB/B′$_{58-72}$-specific TCRs (TCR1 to TCR3) showing the apparent affinity $K_D$ (nM) and the maximum specific binding in units of Dextramer PE mean fluorescence intensity (MFI) (B$_{max}$). TCR1-3 show DR15/ SmB/B′$_{58-72}$ dextramer binding and negative control (Ctrl, dotted gray) shows DR15/ CLIP$_{103-117}$ dextramer that is not expected to bind. **c–h** Image flow cytometry of the interaction of **c** TCR1, **e** TCR2, **g** TCR3 on J76 Jurkats incubated for 2 h with HLA-DR15 B-LCLs in the presence (above) and absence (below) of SmB/B′$_{58-72}$. Arrows point to the mature immune synapse (IS). Quantification of the CD3 and F-actin MFI at the IS of (**d**), TCR1 ($n = 66$ IS cells with SmB/B′$_{58-72}$, $n = 86$ without SmB/B′$_{58-72}$) CD3 $P = < 0.000001$, F-actin $P = 0.000409$, **f** TCR2 ($n = 47$ IS cells with SmB/B′$_{58-72}$, $n = 57$ without SmB/B′$_{58-72}$) CD3 $P = 0.014402$, F-actin $P = 0.000998$, **h** TCR3 ($n = 111$ IS with SmB/B′$_{58-72}$, $n = 83$ without SmB/B′$_{58-72}$) CD3 $P = 0.000720$, F-actin $P = 0.000004$. J76 Jurkats and HLA-DR15 B-LCLs in the presence (red) or absence (brown) of SmB/B′$_{58-72}$. Data are presented as mean with SD. *$P < 0.05$, **$P < 0.01$, ***$P < 0.001$ by Mann–Whitney test. Source data are provided as a Source Data file.

## Smith-specific TCRs are functionally active

Lentiviral vectors encoding TCR1-3 were created with an EF1-alpha promoter controlling the expression of a tricistronic vector encoding the TCR beta chain, TCR alpha chain and eGFP reporter interspaced with P2A and T2A ribosomal skipping sequences (Fig. 6a). The lentiviral vectors were used to transduce J76 Jurkat T-cell line lacking endogenous TCR and primary human Tregs.

We showed that all three SmB/B′$_{58-72}$-specific TCRs can bind to SmB/B′$_{58-72}$-loaded DR15 on B-lymphoblastoid cell lines (B-LCLs) to form an immune synapse, leading to T-cell activation[14]. Hallmarks of HLA-dependent T-cell activation include recruitment of the TCR-CD3 complex to the immune synapse upon TCR binding to HLA[15] and the rearrangement of the cell membrane through the polymerization of filamentous actin (F-actin)[16]. Imaging flow cytometry of J76 Jurkat cells transduced with either TCR1, TCR2 or TCR3 and co-cultured with HLA-

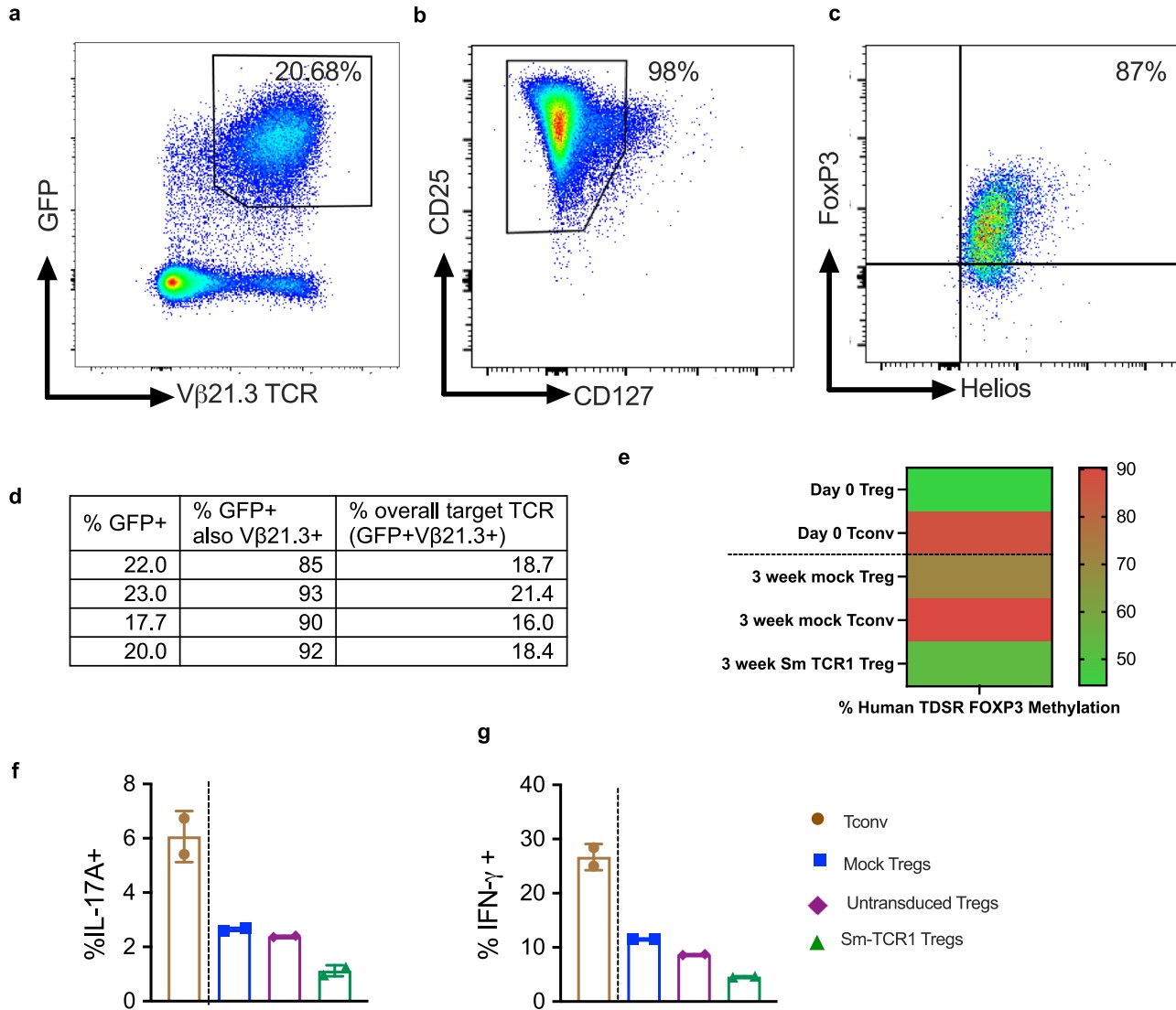

**Fig. 7 | Sm-Tregs are phenotypically stable.** Healthy human Tregs were transduced with TCR1 and expanded 10 days in vitro. **a** Representative flow cytometry dot plot of the CD4$^+$ Treg cells expressing GFP and TCR Vβ21.3, the antibody specific for the variable gene TRBV11-2 of TCR1 showing a mean co-expression of 20.68%. **b** Treg surface marker phenotype by representative dot plots of CD25 and CD127 expression on CD4+ cells. **c** Sm-Treg transcription factor phenotype by representative dot plots of forkhead box P3 (FOXP3) and Helios expression. **d** Treg expression of GFP and Vβ21.3 by flow cytometry from four separate experiments on healthy donor Tregs. **e** methylation of the Treg-cell specific demethylated region (TDSR) of the FOXP3 locus of Treg and Tconv cells at day 0 (before transduction and expansion) and 3 weeks after transduction and expansion. **f, g** IL-17A and IFN-γ expression after stimulation with PMA and ionomycin of Tconv (brown), mock-transduced Tregs (blue), the un-transduced (GFP-) portion of Tregs that underwent lentiviral transduction (purple) and Sm-TCR1-transduced Tregs after 10 days of expansion culture in vitro, measured by intracellular flow cytometry ($n = 2$ biologically independent samples), data are presented as mean with SD. Source data are provided as a Source Data file.

DR15 B-LCLs revealed a significantly greater intensity of CD3ε staining and phalloidin staining, a marker of f-actin polymerization[17], at the immune synapse when cultured in the presence of SmB/B'$_{58-72}$ than in the absence of peptide (Fig. 6c−h). This demonstrates that the ability of SmB/B'$_{58-72}$-specific TCRs to form an immune synapse is increased in the presence of SmB/B'$_{58-72}$ autoantigen, which can result in superior T-cell activation.

Because TCR1 had the greatest apparent affinity for SmB/B'$_{58-72}$-bound HLA-DR15 and induced T-cell activation and memory[18,19], we selected TCR1 for further testing as a potential Sm-Treg therapy.

### Smith T regulatory cells are phenotypically stable

Sm-TCR1 lentiviral vectors were transduced into human Tregs obtained from healthy human donors to determine the multiplicity of infection (MOI), stability of TCR expression and maintenance of

regulatory phenotype. An MOI of 30 was found to be optimal for TCR transduction onto Tregs, resulting in an average transduction efficiency of 20.68% as measured by the percentage of CD4$^+$GFP$^+$TCR Vβ21.3$^+$ cells via flow cytometry (Fig. 7a). TCR1 Treg transduction showed robust co-expression of GFP and TCR as the TCR1 transgene was co-expressed in an average of 90.0% of GFP$^+$ cells confirming the tricistronic vector expressed the transduced genes with appropriately equal stoichiometry (Fig. 7d). Sm-TCR1 Tregs maintained stable transgenic expression of TCR1 long-term in vitro as the expression of target TCR1 remained consistently between a mean expression of 16.68% at day 10 and 13.60% at day 42 (Supplementary Fig. 1a). Sm-Tregs were also found to maintain their regulatory phenotype after in vitro expansion. After 10 days of in vitro expansion culture, 98% of CD4$^+$ Sm-Tregs were found to be CD25$^{hi}$CD127$^{lo}$ (Fig. 7b), and of the Sm-specific Tregs, 87% co-expressed the Treg-specific transcription

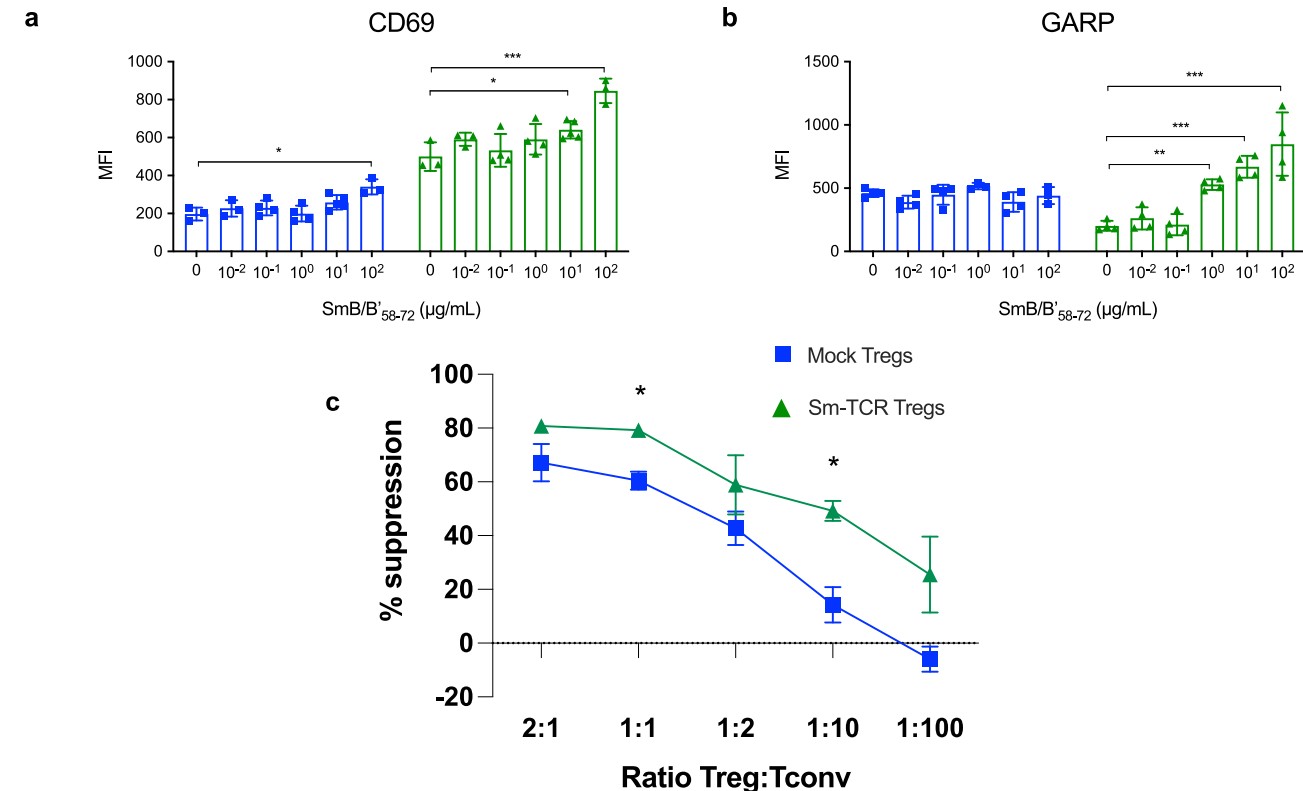

**Fig. 8 | Sm-Tregs are more highly activated and suppress SmB/B'$_{58-72}$ auto-reactivity better than polyclonal Tregs. a** Treg activation measured by CD69 mean fluorescence intensity (MFI) expression by flow cytometry of Tregs transduced with TCR1 (green) or mock-transduced (blue) cultured with B-LCLs pulsed with different concentrations of SmB/B'$_{58-72}$ in triplicate for 36 h. ($n = 3$ (0, $10^{-2}$, $10^2$ condition), $n = 4$ ($10^{-1}$ condition), $n = 5$ ($10^1$ condition) biologically independent samples). **b** Treg activation of the same assay measured by the Treg-specific activation marker, GARP, expression by flow cytometry. ($n = 4$ biologically independent samples except for $n = 3$ ($10^0$ and $10^2$ mock Treg condition). Data are presented as mean with SD. *$P < 0.05$, **$P < 0.01$, ***$P < 0.001$. **c** Suppression assay, measured by proliferation, showing the suppressive capacity of Sm-Tregs on Sm-specific Tconv cells transduced with TCR1 in the presence of SmB/B'$_{58-72}$-pulsed DR15$^+$ B-LCLs and titrations of either TCR1-transduced Tregs (green) or polyclonal mock Tregs (blue). % Suppression is measured by the increase in CellTrace Violet (CTV) MFI from TCR1-transduced Tconvs from the baseline proliferation with no Tregs. ($n = 2$ biologically independent samples) *$P < 0.05$ by unpaired $t$ test. $P = 0.031$ for 1:1 condition and $P = 0.044$ for 1:10 condition. Source data are provided as a Source Data file.

factors Foxp3 and Helios (Fig. 7c) thus confirming the Sm-Tregs are of bona fide regulatory phenotype. Sm-Treg regulatory phenotype stability was further confirmed by methylation analysis of the Treg-cell specific demethylated region (TDSR) of the FOXP3 locus (Fig. 7e). After 3 weeks of in vitro expansion, Sm-Tregs maintained a mean TDSR methylation of 52.8%, consistent with ex vivo Tregs (44.5%) and better than the 3-week-expanded mock Tregs at 70.73% methylated. Conversely ex vivo Tconvs showed a mean TDSR methylation of 87.43%, and 3-week in vitro-expanded Tconvs 90.33% methylation meaning the Sm-Tregs showed consistent TDSR de-methylation long-term, a hallmark of stable Treg phenotype[20]. Concordant with a stable Treg phenotype, Sm-Tregs showed low expression of pro-inflammatory cytokines IL-17A and IFN-γ measured by intracellular flow cytometry (Fig. 7f, g). Artificially stimulated Tregs at 10 days in vitro expansion showed 5.4-fold less IL-17A expression (mean 1.1% IL-17A$^+$) compared to stimulated Tconvs (mean 6.1% IL-17A$^+$). Artificially stimulated Sm-Treg IFN-γ expression was similarly low at 4.6% IFN-γ$^+$ compared to 26.7% IFN-γ$^+$ in stimulated Tconvs. Both IL-17A and IFN-γ expression was lower in Sm-Tregs compared to un-transduced or mock Tregs. Sm-Tregs are therefore able to maintain their suppressor phenotype as there is minimal conversion to pro-inflammatory Th17 cells in vitro.

**Smith Tregs better suppress autoimmunity in vitro**
Following the successful transduction of Sm TCRs onto both J76 Jurkat cells and primary human Tregs, we transduced primary human Tregs from HLA-DR15 healthy donors and performed in vitro co-cultures to assess the suppressive capacity of Sm-Tregs and the specificity of response by serial dilution of SmB/B'$_{58-72}$ peptide. T-cell activation marker CD69 and Treg activation marker Glycoprotein A repetitions predominant (GARP) both exhibited a SmB/B'$_{58-72}$-dependent dose response as measured by flow cytometry. (Fig. 8a, b). The expression of CD69 and GARP was reduced on mock-transduced Tregs. This indicates Sm-Tregs can respond in a dose-dependent manner to SmB/B'$_{58-72}$ presented by HLA-DR15 B-LCLs and not by mock-transduced Tregs expressing a natural TCR repertoire. Based on SmB/B'$_{58-72}$ titration, we selected 100 μg/mL SmB/B'$_{58-72}$ to pulse HLA-DR15$^+$ B-LCLs and co-culture with CD4$^+$ T-conventional cells (Tconvs) transduced with Sm-TCR1 (to increase the population of Tconvs specific for SmB/B'$_{58-72}$) and Tregs, either Sm-TCR1-transduced or mock-transduced. Based on proliferation dye dilution analysis, Sm-Tregs showed heightened inhibition of Sm-Tconv proliferation compared with mock Tregs and no Tregs. A Sm-Treg-dose-dependent suppressive capacity over the Sm-Tconvs was also observed (Fig. 8c). Even at a low ratio of 1:100 Treg:Tconv, the Sm-Tregs showed superior suppressive capacity of 5.33 times more potent suppression of Tconv proliferation than mock Tregs. This indicates that Sm-Tregs are superior to polyclonal Tregs in their suppressive capacity of Sm-Tconvs, although the extent of allo-specific stimulation was unable to be assessed in this assay. Sm-Tregs were also able to show a superior dose-dependent suppressive bystander effect on polyclonal Tconv proliferation compared to mock Tregs (Supplementary Fig. 1b). This shows that in the presence of SmB/B'$_{58-72}$ autoantigen, the Sm-Tregs

**a**

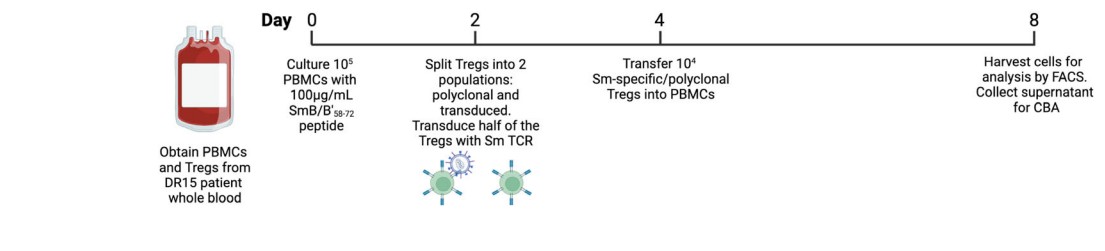

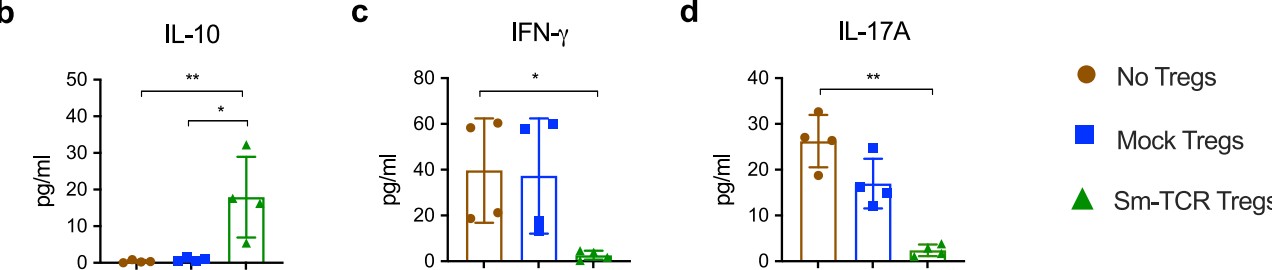

**Fig. 9 | Sm-TCR Treg-dependent in vitro suppression of Sm+ lupus donor PBMC.**
**a** Experimental timeline for co-cultures of PBMCs and Tregs (polyclonal or Sm-specific) obtained from SLE patients with anti-Sm positivity and HLA-DR15. **b** IL-10 concentration in co-culture supernatant measured using cytometric bead array (CBA), No Tregs vs. Sm-Tregs $P = 0.0092$, Mock Tregs vs. Sm-Tregs $P = 0.0111$. **c** IFN-γ concentration in co-culture supernatant measured by CBA, No Tregs vs. Sm-Tregs $P = 0.0140$ and **d** IL-17A concentration in co-culture supernatant as measured by CBA. No Tregs vs. Sm-Tregs $P = 0.0001$. Data are presented as mean with SD ($n = 4$ biologically independent samples). *$P < 0.0332$, **$P < 0.0021$, ***$P < 0.0002$, ****$P < 0.0001$ by ordinary one-way ANOVA and Tukey's multiple comparisons test. Source data are provided as a Source Data file. Created with Biorender.com.

can better suppress Tconvs of other specificities, an effect which contributes to restoring immune tolerance.

Another mechanism Tregs utilize to suppress autoimmunity is through the tolerization of antigen-presenting cells (APCs) such as mature DCs and B cells[21,22]. The tolerizing effects of Tregs acts to reduce the capacity of APCs to present autoantigens to pathogenic T cells through contact-dependent and contact-independent mechanisms. Tregs can also suppress B cells by reducing their ability to secrete autoantibodies and present autoantigens[23,24]. Compared to mock Tregs, Sm-Tregs were able to better tolerize CD19+ B-LCL cell lines as shown by a significant reduction in the expression of HLA-DR, meaning reduced capacity to present autoantigens by HLA as well as a significant reduction in expression of co-stimulatory ligands CD80 and CD86 (Supplementary Fig. 2).

### Smith Tregs suppress SLE patient autoimmunity

To determine if Sm-Tregs would suppress patient autoreactivity and disease in patients with SLE, we transduced the Sm TCR onto Tregs obtained from patients with SLE who were anti-Sm positive and HLA-DR15+, and tested the suppressive function of these Sm-Tregs in vitro and in vivo. For the in vitro analysis, PBMCs were cultured with SmB/B'58-72 for 4 days. Patient polyclonal mock-transduced Tregs or Sm-Tregs were added to the culture and after another 4 days, cytokine concentration in the culture supernatant was measured using a cytometric bead array (CBA) (Fig. 9). This experiment revealed that Sm-Tregs, compared with polyclonal Tregs, were superior at suppressing SLE patient autoreactivity. The concentration of IL-10, one of the signature immunosuppressive cytokines secreted by activated Tregs, was higher in co-culture supernatant with Sm-Tregs than polyclonal Tregs and no Tregs (Fig. 9b). Conversely, concentrations of pro-inflammatory cytokines IFN-γ and IL-17A were significantly lower in co-culture supernatant with Sm-Tregs than polyclonal Tregs and no Tregs (Fig. 9c, d). Elevated levels of these pro-inflammatory cytokines are present in serum and kidney biopsy samples from patients with LN[25,26].

To determine the extent to which SLE patient Sm-Tregs can respond to SmB/B'58-72 antigen stimulation, we co-cultured DR15+ B-LCLs pulsed with SmB/B'58-72 and Sm-Tregs or mock Tregs derived from SLE patients and after culture measured activation markers by flow cytometry. This revealed that Sm-Tregs showed superior activation signals by the pan T-cell activation marker CD69 as well as Treg-specific activation markers GARP and latency-associated peptide (LAP) (Supplementary Fig 3). This shows that despite the reported deficiencies in Treg function in SLE patients, the in vitro culture expansion and Sm-TCR transduction are able to restore patient Treg function and re-direct their responses to target SLE autoantigens.

### Smith Tregs halt lupus nephritis in mice

For the in vivo analysis, we devised a humanized model of LN using NSG-MHC*null* mice. In this model, nephritis is induced by injecting SLE patient-derived PBMCs, and monocyte-derived dendritic cells (DCs) pulsed with SmB/B'58-72 into NSG-MHC*null* mice (Fig. 10a). We show the successful engraftment of human SLE patient-derived Tregs, CD4+ T cells, CD8+ T cells and CD11c+ DCs. Human CD4+ T cells, CD8+ T cells and CD11c+ DCs infiltrate the kidney (Supplementary Fig. 4), and mice start to develop functional renal injury as measured by proteinuria at week 3 and develop significant histological injury as measured by glomerular segmental necrosis, glomerular hypercellularity, mesangial cell proliferation and tubulointerstitial injury at week 8. The NSG mice strain used does not support the long-term engraftment of B cells[27] and as a result antibody responses were not studied. It is therefore hypothesized the kidney injury observed from anti-Sm+ lupus nephritis patient donor cells is driven by T-cell-mediated autoimmunity.

To determine the efficacy of Sm-Tregs in vivo, mice were treated at the onset of functional injury at week 3 with vehicle control (no Tregs), polyclonal Tregs or Sm-Tregs. Mice were culled on day 56 (week 8), and their urine, spleens, and kidneys were obtained for analysis.

Results from this experiment show that patient-derived Sm-Tregs suppressed disease activity significantly better than patient-derived

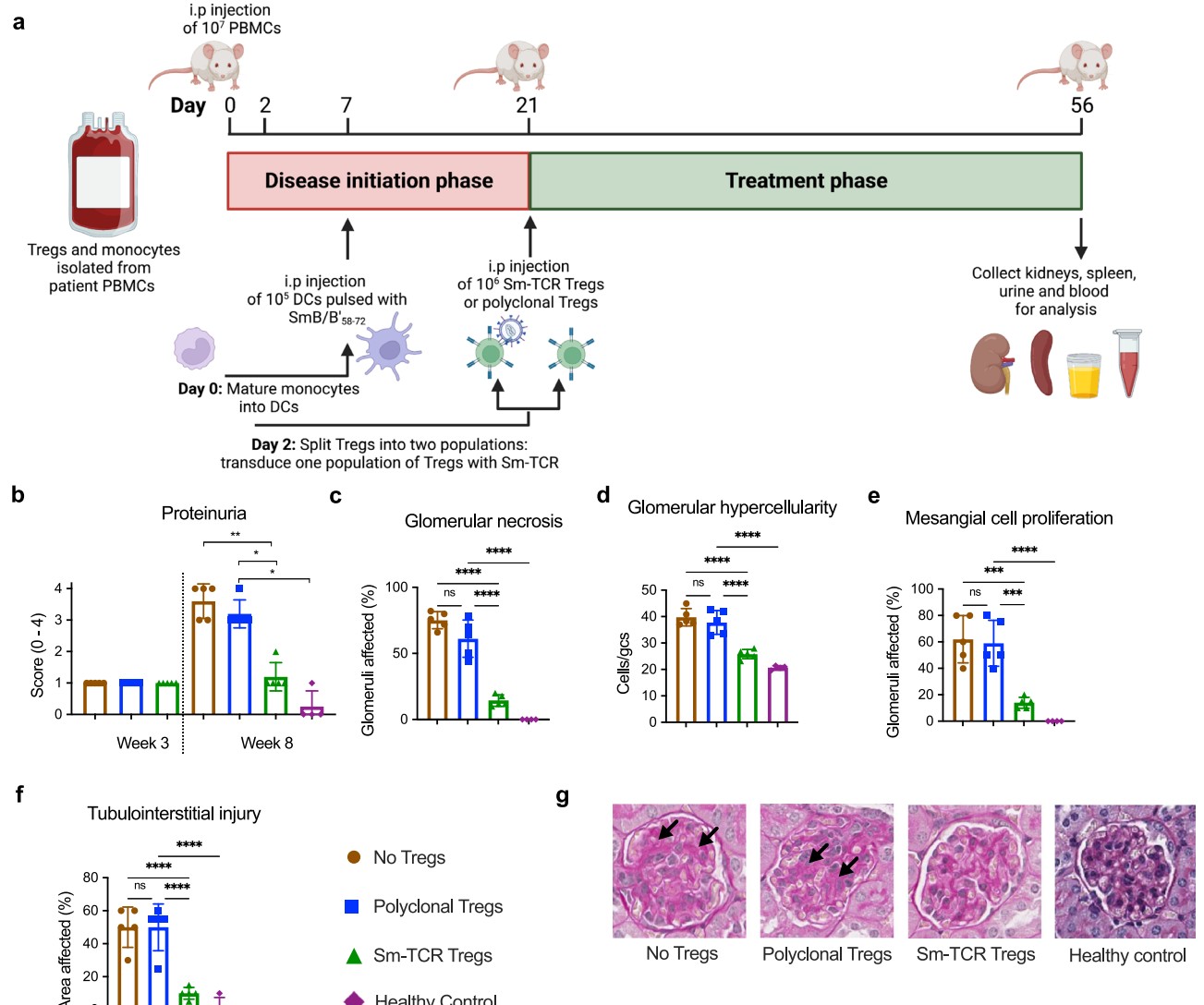

**Fig. 10 | Sm-TCR Treg-dependent in vivo suppression of Sm+ lupus donor PBMC. a** Experimental timeline for in vivo NSG-MHC[null] mouse model of disease. Patient PBMCs, mature DCs pulsed with SmB/B'[58-72] peptide and Tregs (mock polyclonal or Sm-specific) were administered on day 0, day 7, and day 21, respectively. **b** Proteinuria scores measured by urine test strips from mice administered no Tregs (brown), polyclonal Tregs (pTregs, blue), Sm-Tregs (green) or healthy control cells (purple) at week 3 and week 8. **c** Assessments of histological renal injury show the percentage of glomeruli affected by necrosis, **d** glomerular hypercellularity, **e** mesangial cell proliferation, and **f** tubulointerstitial injury from

mice administered no Tregs (brown), polyclonal Tregs (pTregs, blue), Sm-Tregs (green), or healthy control cells (purple) at week 8. **b**–**f** Data are presented as mean with SD. $n = 5$ independent experiments for mice receiving SLE patient cells and $n = 4$ for mice receiving healthy donor cells. *$P < 0.0332$, **$P < 0.0021$, ***$P < 0.0002$, ****$P < 0.0001$ by ordinary one-way ANOVA and Tukey's multiple comparisons test. **g** Representative glomeruli from periodic acid Schiff (PAS)-stained kidney sections from mice administered no Tregs, polyclonal Tregs, Sm-Tregs or healthy control cells at week 8. Black arrows indicate areas of glomerular segmental necrosis. Source data are provided as a Source Data file. Created with Biorender.com.

polyclonal Tregs. At week 3 (before Treg administration), proteinuria scores, a key biomarker for LN[28], were 1+ across all three treatment groups, indicating mild proteinuria and the onset of renal disease (Fig. 10b). At week 8, proteinuria continued to be mild in mice administered Sm-Tregs (mean score = 1.2) whereas proteinuria severity had increased in mice administered polyclonal Tregs (mean score = 3.2) or no Tregs (mean score = 3.6), indicating disease progression. Mice receiving healthy donor cells showed minimal proteinuria (mean score = 0.25) consistent with healthy renal function.

The extent of kidney damage in periodic acid Schiff (PAS)-stained kidney sections reinforced the proteinuria findings; the percentage of necrotized glomeruli was significantly lower in mice administered Sm-Tregs (14.4%) than mice administered polyclonal Tregs (61.2%) or no Tregs (75.2%) (Fig. 10c). The extent of glomerular hypercellularity was

significantly lower in mice administered Sm-Tregs (25.8 cells/glomerular cross-section (gcs)) compared to those administered polyclonal Tregs (37.8 cells/gcs) or no Tregs (39.8 cells/gcs) (Fig. 10d). Mesangial cell proliferation was significantly reduced in Sm-Treg administered mice (14% glomeruli affected) compared to polyclonal Tregs (59%) or no Tregs (62%) (Fig. 10e). Tubulointerstitial injury was also significantly reduced in mice receiving Sm-Tregs (10% area affected) compared to polyclonal Tregs (50%) or no Tregs (50%) (Fig. 10f). Glomerular crescents were not observed in any sample. To control for graft–versus–host disease (GVHD), a group of mice that received healthy donor cells and un-transduced Tregs showed no signs of weight loss or renal disease that could be attributed to GVHD (Fig. 10b–g), consistent with previous reports using this NSG mouse strain[27].

## Discussion

In this study, we have successfully identified HLA-DR15-restricted, Sm-specific, high-affinity TCRs using a method of physical binding assays, DC and PBMC co-cultures, and high-throughput single-cell sequencing. We have also shown that these Sm-specific TCRs, once transduced onto primary patient Tregs, have conferred Sm-specificity and the ability to functionally suppress Sm-autoreactivity both in vitro and in vivo. In addition to this main finding, there are a few points we would like to highlight.

The protein crystal structure of SmB/B'$_{58-72}$ in complex with HLA-DR15 proves SmB/B'$_{58-72}$ binds HLA-DR15, defines its interaction and confirms the nature of the structure is indeed an antigen-specific interaction that binds in a conventional manner. This structure is one of the only three peptide-HLA structures available to date for human self-antigens in the context of HLA-DR15. Knowledge of the structural characteristics defining which autoantigenic peptide residues bind to HLA-DR15 will assist with the design and improvement future candidate TCRs for lupus nephritis as well as other HLA-linked disease conditions. This structural analysis will facilitate future off-target screening as key structural features of the autoantigen that can be recognized by Sm TCRs have been determined.

Although scRNASeq data were generated on in vitro-expanded CD4$^+$ T cells and as such does not reflect the in vivo expression profile of the precursor cells, it is worthwhile to assess the T-cell subtypes giving rise to the dominant TCR subpopulations. Interestingly, the significant upregulation of *CD52* in cells expressing TCR1 suggests that TCR1 is associated with a type of nonconventional suppressor T cell based on previous research in type 1 diabetes (T1D) showing that CD4$^+$CD52$^{hi}$ cells possess suppressive properties[29]. Other significantly upregulated genes on the TCR1$^+$ subset were also associated with potential suppressive properties, such as *LAIR2* and *IL9R*[30,31].

The Sm-Treg in vivo mouse model did not receive any IL-2 adjunct therapy, as previously reported clinical trials of Tregs in autoimmune diseases did not supplement IL-2 after transfer and Tregs remained in the patients for up to 1 year after transfer, suggesting IL-2 supplementation is not necessary[4]. Trials have looked at using low-dose IL-2 to restore Treg function in SLE[32]. Such trials aim to address Treg and IL-2 signaling dysfunction of SLE patients[33]. A previous report of Treg therapy in humans showed the Treg in vitro expansion with high-dose IL-2 prior to transfer, a similar method to ours, can restore donor Treg function in T1D by increasing signal transducer and activator of transcription 5 (pSTAT5), a marker of IL-2-dependent Treg function[4]. Given the in vitro supplementation of IL-2 is sufficient to restore Treg function and promote long-term engraftment, we deemed IL-2 adjunct therapy unnecessary for Sm-Tregs.

In conclusion, using LN as a proof of principle, we have developed a platform to establish TCR Treg-cell-based therapy that can be extended into other manifestations of SLE, including targeting different autoantigens and HLA-types as well as in other HLA-linked autoimmune diseases. We aim to further characterize Sm-Tregs in our humanized LN mouse model by tracking Sm-Tregs long-term in vivo and assessing their persistence and phenotype. The Sm-Treg production method is being developed into a GMP-ready, closed-system method of manufacture, which can then be used as a cell-based therapeutic in a phase I clinical trial for patients with LN.

## Methods

### Sm epitope identification

A biophysical affinity binding assay, MHC Class II Reveal (Proimmune), was used to determine HLA-DRB1*15:01 (HLA-DR15)-restricted T-cell epitopes[34]. A total of 145 15-mer peptides offset by three amino acids from SmB/B' (also known as SNRPB, GenBank Accession: AAA36578.1, AAA36579.1), SmD1 (also known as SNRPD1, GenBank Accession AAH72427.1) and SmD3 (also known as SNRPD3, GenBank Accession AAH03150.1) were screened against HLA-DR15 at 37 °C. If the peptide

bound to HLA-DR15, a specific monoclonal antibody bound and a fluorescent signal were released. Fluorescent signal measurements of peptide-HLA binding were taken at 0 h and 24 h and given as Reveal scores, defined as peptide-HLA binding relative to a positive control peptide, which is a known T-cell epitope for HLA-DR15. The Reveal scores were used to derive a half-life approximation based on a one-phase dissociation equation. Specifically, the equation to fit data to generate off-rates was $Y = A1 * EXP(-A2 * X)$ where $Y$ = the Reveal Score at 24 h, $A1$ = the $Y[0]$ also known as the Reveal Score at 0 h and $A2 = k$ also known as the rate constant. The rate constant, $k$, was defined as $k = -((\ln(Y/Y[0]))/X)$, where $X$ = time in hours. Then the rate constant ($k$) was used to calculate the on- and off-rate half-life values ($t_{1/2}$) according to the equation $t_{1/2} = \ln2/k$. For analysis, a stability index score was used as a normalized measure of peptide-MHC binding by the following equation: Stability Index = $t_{1/2} * Y[0]/100$. The Stability Index was used to weight the value to minimize the over-representation of weakly binding peptides. For subsequent analysis, 15-mers were synthesized with a N-terminus (H-) and a C-terminus (-OH) (Mimotopes). Peptide purity was determined as ≥90% by reversed-phase high-performance liquid chromatography. Peptides were reconstituted at 5 mg/mL in 5% dimethyl sulfoxide (DMSO, Sigma) in sterile water (v/v).

### Inclusion and ethics

All procedures were performed in accordance with the Declaration of Helsinki. All donors provided written informed consent. Lupus patient samples and data were obtained through the Australian Lupus Registry and Biobank (ALRB)[35]. This study was approved by the Human Research Ethics Committees from Monash Health (Project ID 36506) and Monash University (Project ID 18277). All animal study procedures were reviewed and approved by the Monash Animal Ethics Committee (Project ID MMCB-2019-16-Ooi).

### Sm epitope immunogenicity

The top three Sm epitopes were tested in vitro using whole blood obtained from a healthy adult male homozygous for HLA-DR15. Thirty mL of whole venous blood was obtained in EDTA Vacutainers (BD). Mononuclear cells (MNCs) were isolated with Lymphoprep in SepMate tubes (StemCell). Monocyte isolation was then performed using EasySep Human Monocyte Isolation Kit (StemCell). Isolated monocytes were stained with the proliferation dye CellTrace Far-Red (CTFR, Invitrogen), and differentiated into mature dendritic cells with ImmunoCult DC Differentiation Kit (StemCell).

Seven days later, another 30 mL of autologous blood was obtained. CD4$^+$ T cells were isolated using RosetteSep Human CD4$^+$ T-Cell Enrichment Kit (Stemcell). CD4$^+$ T cells were labeled with Cell-Trace Violet (CTV, Invitrogen), resuspended in supplemented RPMI (10% human male AB serum, 2% penicillin/streptomycin, 2 mM ʟ-glutamine and 50 μM 2-mercaptoethanol) and co-cultured with DCs at a ratio of 1 CD4$^+$ T cell to 1 DC. Co-cultures were supplemented with 100 μg/mL Sm peptide and 80 units/mL recombinant-human IL-2 (StemCell). Cells were cultured at 37 °C, 5% CO$_2$ for 6 days then harvested and stained with anti-human antibodies; CD4 BUV496 (clone SK3, BD), CD8 APC-H7 (clone SK1, BD), CD69 BV711 (clone FN50, BD), HLA-DR AF488 (clone L243, Biolegend) and propidium iodide (PI) (Sigma) before flow cytometry analysis using an LSR Fortessa X20 (BD).

### Expression and purification of HLA-DRB1*15:01

The cDNA encoding extracellular domains of HLA-DR15 α- and β-chains were separately cloned into the pHLsec expression vector (Addgene plasmid #99845). The construct contained C-terminal enterokinase cleavable fos/jun zippers to promote dimerization. The β-chain contained a BirA site for biotinylation and a Histidine tag for IMAC purification on the C-terminus. HLA-DR15 was expressed with the SmB/

$B'_{58-72}$ covalently attached via a Factor Xa cleavable linker to the N-terminus of the β-chain and is preceded by a Strep-II tag (IBA) for purification. In brief, the HLA-DR15 construct was transiently expressed in Expi293F (GnTI−/−) cells (kind gift from the Whisstock Laboratory, Monash University), and soluble protein was harvested from culture supernatant. The soluble protein was dialyzed into 10 mM Tris pH 8.0, 500 mM NaCl and purified using immobilized metal affinity (HisTrap™ High Performance; Cytiva) and size-exclusion (Superdex 200; Cytiva) chromatography in 10 mM Tris, pH 8.0, 150 mM NaCl.

## Protein crystallization and structural determination
For protein crystallization, HLA-DRB1*15:01_ SmB/B'$_{58-72}$ was buffer-exchanged to 25 mM Tris, pH 7.6, 50 mM NaCl, and 2 mM $CaCl_2$, followed by removal of the fos/jun zippers with enterokinase (New England Biolabs) at room temperature for 16 h. The extracellular domain of the HLA-DRB1*15:01_SmB/B'$_{58-72}$ was purified using anion-exchange chromatography (HiTrap Q HP; Cytiva) and concentrated to 3.5 mg/ml for crystallization. HLA-DRB1*15:01_ SmB/B'$_{58-72}$ was crystallized in the presence of 20% PEG3350, 200 mM sodium malonate pH 4.0. Diamond-like crystals were exposed to cryoprotectant, 20% glycerol for 30 s before being flash-cooled in liquid nitrogen. Datasets were collected on the MX2 beamline of the Australian Synchrotron. Data were integrated with XDS and scaled and merged in Aimless. Phases were obtained using molecular replacement in PhaserMR, PHENIX[36]. The PDB entry 6CPO (chain A and B, without glycans or peptides) was used as search model for HLA-DRB1*15:01 for molecular replacement. The structures were built and refined in Coot and PHENIX[36]. The final model was validated in MolProbity[37]. Crystallographic data were uploaded to Worldwide Protein Data Bank accession 8TBP, and a crystallographic data summary is found in Supplementary Table 2.

## 10X single-cell immune profiling
CD4$^+$ T cells were cultured with SmB/B'$_{58-72}$-pulsed DCs as per Sm epitope immunogenicity in the methods above. Post-staining, CD4$^+$CTV$^{lo}$ cells were sorted using an Aria cell sorter (BD) and concentration adjusted to 0.7–1.2 × 10$^6$ cells/mL in 0.04% BSA/PBS before loading into 10x Chromium Controller (10X Genomics) for single-cell RNASeq preparation following the instructions of the manufacturer (10X Genomics) at Micromon Genomics. Gene expression and V(D)J libraries were sequenced on a NextSeq System (Illumina), and single-cell TCR sequencing data were prepared in Cell Ranger (10X Genomics, version 3.0.2) followed by visualization and analysis on Loupe Cell Browser (10X Genomics, version 4.2.0) and Loupe VDJ Browser (10X Genomics, version 3.0.0). ScRNASeq data is uploaded to Gene Expression Omnibus accession number GSE242152.

## Sm TCR lentiviral transfer plasmid generation
Plasmids containing the SmB/B'$_{58-72}$-specific TCR1-3 were synthesized (GeneArt). Each TCR insert encoded 5′ to 3′ the TCR beta chain, followed by P2A from porcine teschovirus-1 2 A, followed by the TCR alpha chain, followed by T2A from thosea asigna virus 2A, followed by the eGFP reporter all under the control of an EF1 alpha promoter.

Plasmids were transformed into One-Shot Top10 chemically competent *E.coli* (Invitrogen). Genes of interest were digested with XbaI and EcoRI-HF (New England Biolabs) and cloned into lentiviral backbone vector (Creative Biolabs) using One-Shot STBL3 chemically competent cells (Invitrogen). Correct insertion of TCR1-3 into lentiviral vectors were confirmed via Sanger sequencing (Micromon Genomics).

## Lentiviral vector preparation
Lentiviral vector encoding our Sm TCRs of interest were generated using the Gibco LV-MAX lentiviral production system (ThermoFisher). Gibco viral producer cells were transfected using pMD2.G, pRSV.rev, pMDL/pRRL (Addgene), and transfer plasmids at a molar ratio of 1:2:1:4. After 48 h, supernatant was collected and filtered through a 0.45 μm low protein-binding PVDF vacuum-driven filter (Merck). Lentiviral vectors were then concentrated using sucrose gradient ultra-centrifugation with an underlay of sucrose buffer at 19,000 rpm for 2 h at 4 °C. After ultracentrifugation, viral pellet was resuspended in ice-cold HBSS and stored at −80 °C. Functional titration of lentiviral vectors measured in TU/mL was performed on J76 Jurkat cells (kind gift from Mirjam Heemskerk, Leiden University Medical Center)[38].

## Sm-TCR Jurkat cell lines
Jurkat J76 cells lacking endogenous TCR were lentivirally transduced with either TCR1, TCR2, or TCR3 and then GFP + PI- cells were sorted on a FACS Aria (BD)[38]. The sorted cells were then expanded in vitro in supplemented RPMI and used for apparent affinity determination and imaging flow cytometry.

## Apparent affinity determination of SmB/B'$_{58-72}$-specific TCRs
10$^5$ TCR-transduced J76 cells were stained with DRB1*1501/RVLGLVLLRGENLVS/PE dCODE dextramer according to the manufacturer (Immudex) with the following amendment; Instead of 1 test being used for dextramer staining, cells were stained with seven two-fold serial dilutions of dextramer starting at 5 nM, plus 0 nM. After which, cells were stained with 2 μL PI before acquisition of 10,000 events per sample on an LSR Fortessa X20 flow cytometer (BD). Samples were analyzed in FlowJo 10.6.2 and the MFI values of the singlet, alive cells in the Dextramer PE channel were used in the apparent affinity calculation. Apparent affinity, measured in $K_D$, was derived from the negative reciprocal of the slope of the line of fit to Scatchard plots of bound dextramer/free dextramer (mean fluorescence intensity (MFI)/nM) versus bound dextramer (MFI), as described previously[39].

## Cell preparation for imaging flow cytometry
B-lymphoblastoid cell line (B-LCL) homozygous for HLA-DR15 was prepared, as described[40] and pulsed with 100 μg/mL SmB/B'$_{58-72}$ in serum-free RPMI for 2 h at 37 °C. 10$^6$ Sm-TCR1, TCR2, or TCR3 J76 Jurkats and 10$^6$ pulsed B-LCLs were mixed in 200 μL serum-free RPMI and incubated at 37 °C for 1 h then fixed with 1× fixation buffer (eBioscience) at 4 °C for 8 h, washed in 1× permeabilization buffer (eBioscience) and then resuspended in 200 μL antibody/phalloidin cocktail containing 1.25 μL anti-HLA-DR BV421 (clone G46-6, BD), 2.5 μL anti-CD3ε PE (clone OKT3, Invitrogen) and 0.7 μL 400× phalloidin AF647 (Invitrogen) in 1× permeabilization buffer for 45 min. Cells were washed and resuspended in PBS for imaging flow cytometry.

## Imaging flow cytometry
In total, 1 μg/mL PI was added to samples immediately prior to acquisition. A minimum of 30,000 events was acquired for each sample on the Amnis Imagestream X Mark II (Luminex) using Inspire software with a 60X objective, calibration beads, a 7-μm core stream diameter and lasers set at 50 mW (405 nm laser), 140 mW (488 nm) and 80 mW (642 nm). Analysis was performed using Ideas Software ver.6.2 (Luminex). Batch analysis, converting the raw image files (.rif) to data analysis files (.daf) files, was performed on all samples using the above strategy to ensure the same settings were applied to each sample. The gating and mask strategy is found in Supplementary Fig. 5. Mean pixel intensity data of CD3 and F-actin at the immune synapse was analyzed in GraphPad Prism (version 8.3.1).

## Sm-Treg production
PBMCs from healthy human buffy coats (Australian Red Cross Lifeblood) were isolated using Lymphoprep and SepMate tubes. PBMCs underwent a red blood cell lysis step with ACK lysing buffer (ThermoFisher) and then were counted on a hemocytometer. PBMCs were used fresh, or frozen and thawed at a later date. EasySep Human CD4$^+$

T-cell Isolation Kit followed by EasySep Human CD4+CD127lowCD25+ Regulatory T Cell Isolation Kit (Stemcell) was used to isolate Tregs following the steps by the manufacturer. Tregs were plated at 50,000 cells in 200 μL of StemCell ImmunoCult-XF T cell Expansion media supplemented with 10% male human AB serum plus ImmunoCult Human CD3/CD28/CD2 T-cell Activator and 1000 IU/mL IL-2 in a 96-well plate. Three days after Treg isolation, they were lentivirally transduced by spinoculation at an MOI 30 with Lentiboost and Protamine Sulfate (8 μg/mL) at 31 °C for 2 h at 1500×$g$ then incubated at 37 °C. The next day, media was exchanged with fresh media (as above) and re-incubated. Cultured Tregs were supplemented with 1000 IU/mL IL-2 every 2 days.

## Sm-Treg phenotype analysis

In vitro-expanded Tregs were stained with Live/Dead Aqua (Life Technologies) followed by the following anti-human antibodies; CD4 BUV496 (clone SK3, BD), CD127 APC-Vio770 (clone REA614, Miltenyi), CD25 APC (clone BC96, Biolegend) and TCR Vbeta21.3 PE (clone REA894, Miltenyi).

For intracellular staining: Tregs were stimulated with PMA/ionomycin and Brefeldin A (Sigma) for 4 h 37 °C after which cells were fixed and permeabilized with Foxp3/Transcription Factor Staining Buffer Set (eBioscience) and stained with anti-human antibodies; Foxp3 (clone 236A/E7, BD), Helios (clone 22F6, Biolegend), IFN-γ (clone B27, BD) and IL-17A (clone N49-653, BD). Cells were acquired on a Cytek Aurora 5 L flow cytometer and analysis performed in FlowJo (version 10.6.2). A gating strategy is found in Supplementary Fig. 7. Methylation of the FOXP3 TDSR was performed by pyrosequencing of intron 1 of TDSR region -2330 to -2263 from ATG of FOXP3 and analysis of 9 CpG sites by EpigenDx.

## In vitro peptide dose–response assay

For SmB/B'58-72 peptide titration, a co-culture of 10^5 TCR1-transduced Tregs or 10^5 mock-transduced Tregs and 50,000 DR15 + B-LCLs pulsed with SmB/B'58-72 in five tenfold serial dilutions from 100 μg/mL to 0.01 μg/mL and no peptide was established. Each condition was plated on a 96-well U bottom plate in supplemented RPMI and incubated in a 37 °C, 5% $CO_2$ humidified incubator for 36 h after which cells were harvested for flow cytometry. Cells were first stained with Live/Dead Blue (Invitrogen) followed by surface marker staining with anti-human antibodies; CD4 R718 (clone SK3, BD), TCR Vbeta21.3 PE (clone REA894, Miltenyi), CD69 BUV395 (clone FN50, BD) and GARP BV786 (clone 7B11, BD). Cells were acquired on a LSR Fortessa X20 flow cytometer (BD) and analyzed in FlowJo 10.6.2. MFI values were exported and analyzed in GraphPad Prism ver.8.3.1.

## In vitro suppression assay

TCR1-transduced Tconvs (10^5) were stained with CTV and co-cultured with 50,000 DR15+ B-LCLs pulsed with 100μg/mL SmB/B'58-72 and serial dilutions of either TCR1-transduced Tregs stained with CTFR or mock-transduced Tregs stained with CTFR. Each condition was plated on a 96-well U bottom plate in 200 μL supplemented RPMI and incubated in a 37 °C, 5% $CO_2$ humidified incubator for 5 days after which cells were harvested for flow cytometry. Cells were first stained with Live/Dead Near Infrared cell viability stain (Invitrogen) followed by surface marker staining with anti-human antibodies; CD3 PerCP (clone SK7, Biolegend), TCR Vbeta21.3 PE (clone REA894, Miltenyi) and CD19 PE-CF594 (clone HI519, BD). The entire sample was acquired on a Cytek Aurora 5 L flow cytometer and analyzed in FlowJo 10.6.2. The gating strategy is found in Supplementary Fig. 8. The Tregs and Tconvs were distinguished by CTV and CTFR dye labels. The % suppression was calculated by: % proliferation of Tconv's alone minus % proliferation of Tconv's with Tregs/% proliferation of Tconv's alone ×100. Graphs were constructed in GraphPad Prism ver.8.3.1.

## In vitro patient cultures

Adult SLE patients who fulfilled the American College of Rheumatology Classification criteria for SLE and had anti-Sm autoantibodies, and positive for HLA-DR15 had whole blood drawn and PBMCs isolated with Lymphoprep in SepMate tubes. The resulting PBMCs were FACS-sorted into four groups on the basis of CD4, CD8, CD25, CD127 and CD14. These four groups were (1) CD4+CD8-CD25hiCD127lo Tregs; (2) CD4+CD8-CD25loCD127hi Tconvs; (3) CD8+CD4+CD14+ monocytes and (4) CD8+CD4+CD14- PBMCs (Supplementary Fig. 6). Surface staining for the sort was performed using anti-human antibodies; CD4 Pacific Blue (clone RPA-T4, Biolegend), CD8 AF488 (clone HIT8a, Biolegend), CD127 PE (clone A019D5, Biolegend), CD25 BUV395 (clone 2A3, BD), CD14 APC-Cy7 (clone MOP9, BD) and PI. Post-sort, cells from groups 2 and 4 were combined and stained with CTV and cultured with 100 μg/mL SmB/B'58-72 peptide (Mimotopes). Monocytes from group 3 were differentiated into DCs using the ImmunoCult DC culture kit (StemCell), and matured DCs were used for in vivo experiments. To quantify the secretion of IL-10, IFN-γ, and IL-17A, co-culture supernatants were harvested and a cytometric bead array (CBA) (BD) was used.

## Patient Treg in vitro activation assay

Patient-derived TCR1-transduced Tregs (10^5) were stained with CTV and co-cultured with 50,000 DR15+ B-LCLs pulsed with 100 μg/mL SmB/B'58-72 and 10^5 polyclonal Tconv's. Cells were plated on a 96-well U bottom plate in 200 μL supplemented RPMI and incubated in a 37 °C, 5% $CO_2$ humidified incubator for 5 days, after which cells were harvested for flow cytometry. Cells were stained with Live/Dead Aqua stain (Invitrogen) followed by surface marker staining with anti-human antibodies; CD3 PerCP (clone SK7, Biolegend), LAP PE-Cy7 (clone FNLAP, Invitrogen), GARP (clone BB700, BD) and CD69 BV711 (clone FN50, BD). Samples were acquired on a Cytek Aurora 5 L flow cytometer and analyzed in FlowJo 10.6.2.

## In vivo humanized NSG mouse model of lupus nephritis

Mice used in experiments were NSG mice which have mouse MHC class II deleted (NOD.Cg-Prkdcscid H2-Ab1em1Mvw H2-K1tm1Bpe H2-D1tm1Bpe Il2rgtm1Wjl/SzJ, Jackson Laboratories strain 025216)[27]. Sex was not considered in the study design. Mice were assigned randomly to groups. Patient PBMCs, monocytes, and Tregs were obtained from the same four-way sort as described for in vitro co-cultures. Treg-depleted PBMCs (10^7) in PBS were injected into 8–10-week-old naive NSG mice intraperitoneally on day 0. Monocytes were differentiated into DCs, as described above. On day 7 of culture, 10^6 matured DCs were pulsed with 100 μg/mL of SmB/B'58-72 peptide for 1 h in PBS, washed, and injected into mice intraperitoneally in PBS. Tregs were transduced with Sm-TCR1 as described above and expanded in vitro. Cultures were expanded with anti-CD3/CD28 microbeads and 300 IU/mL IL-2 was added every two days. On day 21, 10^6 mock-transduced (polyclonal) or Sm-Tregs were harvested and injected intraperitoneally into mice in PBS.

Mice were placed in metabolic cages for the collection of urine 14 h before euthanasia. Proteinuria was measured using a urine dipstick (Siemens Multistix) on day 21 prior to Treg administration and at day 56. Proteinuria was scored as 0 (negative or trace), 1 (0.30 g/L), 2 (1 g/L), 3 (3 g/L) or 4 (≥20 g/L) according to dipsticks.

Mice were humanely euthanized on day 56 via $CO_2$ asphyxiation. Mouse kidneys were harvested for histological assessment.

## Histological assessments

Kidney injury was assessed on formalin-fixed, paraffin-embedded, periodic acid Schiff (PAS)-stained kidney sections, as previously reported[41]. Glomerular segmental necrosis was scored as an accumulation of ≥50% PAS-positive material in the glomerulus, and glomerular crescents as two or more layers of cells in the Bowman's space. Glomerular hypercellularity was scored as the number of cells per

(glomerular cross-section) gcs in at least 20 consecutive equatorial glomeruli. Mesangial cell proliferation was scored on at least 20 consecutive equatorial glomeruli as % glomeruli affected, which is defined as a glomerulus with three or more nuclei clumped together. The tubulointerstitial injury was assessed by giving a score of 1 per high-powered field (hpf) if any of the following were present: tubulointerstitial infiltrates, tubular dilation, necrosis, or protein cast formation (min. 20 hpf were counted).

Immunofluorescent staining was performed on 5 µm snap frozen kidney sections using primary antibodies goat anti-human CD4 (AF379; R&D Systems); rabbit anti-human CD8 (NBP2-29475; Novus Biologicals); and Armenian hamster anti-human CD11c (NB110-97871; Novus Biologicals) and Alexa Fluor-conjugated secondary antibodies (705-545-003, 711-585-152, 127-605-160; Jackson ImmunoLab). DAPI (D1306, ThermoFisher Scientific) was used for nuclear staining. Images were scanned by VS120 slide scanner (Olympus Life Science) using the OlyVIA Software (Olympus Life Science), and images were taken using QuPath (https://qupath.github.io/; University of Edinburgh).

## Statistical analyses
Values are presented as mean ± SD; *$P < 0.05$, **$P < 0.01$, ***$P < 0.001$, ***$P < 0.0001$ as measured by either an unpaired $t$ test when measuring comparing between two groups or a Kruskal–Wallis test or one-way ANOVA when comparing between more than two groups. Single-cell sequencing data $P$ values are derived from a negative binomial exact test with adjustment using Benjamin Hochberg correction for multiple tests.

## Reporting summary
Further information on research design is available in the Nature Portfolio Reporting Summary linked to this article.

## Data availability
ScRNASeq data is uploaded to Gene Expression Omnibus accession number GSE242152. Crystallographic data was uploaded to Worldwide Protein Data Bank accession 8TBP and a crystallographic data summary is found in Supplementary Table 2. Source data are provided with this paper.

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

## Acknowledgements

The authors wish to thank all donors for their blood donations: Angela Vais from the Monash Histology Platform for histology work, Michael Thomson from the Monash Health Translation Precinct (MHTP) Flow Core Facility for flow cytometry sorting, the staff at Micromon Genomics for single-cell sequencing and staff at MHTP Animal Facilities for mouse colony care and management. Relevant funding was awarded by Lupus Research Alliance Inc., Funder ID 100012051, 588087 and 850279 (J.D.O.), National Health and Medical Research Council (NHMRC), Funder ID 501100000925, 2017877 (J.D.O.), Amgen Inc., Funder ID 100002429, 2021026213-001 (J.D.O.), United States Department of Defense, Funder ID 100000005, LR210065 (J.D.O.).

## Author contributions

P.J.E, R.M.C., J.D.O., E.F.M., S.L.S. and Y.T.T. designed the research and wrote the paper. J.A.M. wrote the paper. P.J.E, R.M.C., C.L., J.C., B.H.N., Y.T.T., K.L.L., A.B., E.S.T., S.L., C.S. and J.X.C. performed and analyzed experiments. E.F.M., R.K.R., R.K. and A.H. provided samples.

## Competing interests

The authors declare the following competing interests: Relevant funding was awarded by Lupus Research Alliance Inc., Funder ID 100012051, 588087, and 850279 (J.D.O.), National Health and Medical Research Council (NHMRC), Funder ID 501100000925, 2017877 (J.D.O.), Amgen Inc., Funder ID 100002429, 2021026213-001 (J.D.O.), United States Department of Defense, Funder id 100000005, LR210065 (J.D.O.). The research has been partially supported by Amgen. PCT patent application (PCT/AU2021/050254) has been filed by Monash University as the sole applicant. Inventors: J.D.O., P.J.E., and E.F.M. The remaining authors declare no competing interests.
