## [Peer Review File · Nature Communications]

Smith-specific regulatory T cells halt the progression of lupus nephritisREVIEWER COMMENTS

Reviewer #1 (Remarks to the Author):

Cheong et al. report the design of Sm-TCR specific Tregs for HLA-DRB1:15 positive individuals and demonstrate that they can suppress T cell activation in vitro and kidney damage in vivo in a humanized model of lupus nephritis. They first identify 15mer peptide fragments that span Sm proteins to identify those that bind with highest binding % and affinity to HLA-DRB1:15. They then identify the crystal structure of HLA-DRB1:15 interacting with the top peptide (SmB/B'58-72). They use TCR sequencing and co-culture studies to identify a TCR clonotype that recognizes this complex with high affinity (TCR1), and then they incorporate the TCR alpha and beta chains for this TCR into a lentivirus that they can then transduce conventional and regulatory T cells with to conduct functional studies. The in vivo model used is an MHC null immunodeficient mouse that has been provided with patient PBMCs and later mo-DCs with Sm antigen provided. Mice were treated at week 3 with autologous Sm-TCR specific Tregs and readouts were at week 3 and week 8.

Strengths of this paper include the technological sophistication and the logical step-wise approach to designing antigen specific Tregs and testing them in vivo. It is also logistically challenging to use patient cells, and this is an elegant addition to the paper.

However, I was surprised by how many figures were put in the Supplemental and wonder about bolstering the main text with additional data. I also think the Methods could be condensed - some descriptions could be removed for basic lab protocols or repeated steps (e.g. incubator conditions, mentioning how blood is obtained for each experiment). By shortening the Methods, this could allow greater word count availability to frame the work within the context of the field.

Figure 2: I wonder if ab responses were measured and whether anti-Sm was specifically suppressed. This point should pin-point the claimed specificity. Many other parameters of kidney injury are not mentioned or discussed. For example, cell infiltration in the glomeruli, mesangial cell proliferation, I assume there were no crescents, Tubular epithelial damage, interstitial inflammation and so on.

Some questions that I have, for example, include whether the authors think that the Tregs are predominantly acting through contact dependent vs. independent manners. I think it is clear that the Treg cells are activated by a specific antigen – this is by design. However, are they recognizing the Sm peptide and acting by bystander suppression, are they blocking Tcon from binding to the MHC-peptide (especially TCR1+ Tcon), are they killing activated T cells, and/or are they otherwise exerting their regulatory effect on Tcon/APCs? This matters because if framed as being antigen specific, the question arises as to how they would behave in human lupus in which there are many auto-antigens present. The in vivo model is purely based on Sm antigen and does not test this. Meanwhile, if Sm peptides are used to activate Treg that act by bystander suppression, this might cover the more complex autoimmune situation in patients, but then it would be a broader type of immune suppression (not antigen specific). Either way would potentially be useful, so this is not a weakness but rather something that could be discussed. This approach implies that such Sm-specific Tregs will be used in Sm positive lupus patients under the assumption that these abs account for or are involved in disease pathogenesis. And a question to discuss, is how they propose to choose autoantigens beyond Sm.

Not directly relevant to this paper but worth mentioning for translational considerations is the short half-life of Treg in vivo and the IL-2 deficient state of patients with SLE that may lead to an even shorter half-life. I noticed the IL-2 dose used in culture is 1000IU/mL for culture from healthy buffy coat and 300IU/mL for those isolated from lupus patients. Is the reason for this related to the second scenario not being an expansion phase? Would the use of lower dose go against the fact that IL-2 production is low in SLE, and, more importantly, that T cells from patients with SLE do not generate sufficient amounts of pSTAT5 in response to IL-2? Given the flurry of efforts to treat patients with lupus with low dose IL-2, should Ag-specific Tregs be used in conjunction with IL-2? – a discussion point.

Text Comments

Abstract line 24 – perhaps it is meant to say “our lead”?

Page 3 line 37 – ‘high mortality’ is a bit vague; it may be more accurate to say that ‘LN is an important contributor to disease related morbidity and mortality’

Page 3 line 41 – is there a word missing after ‘Sm antigen-specific’?

Page 3 line 51 – could include parenthetical statement regarding the meaning of ‘stability index’, either in text or the figure legend; I see how the calculation is made in the Methods, but perhaps a brief descriptive statement in the main text could help the reader understand the conclusion

Page 4 – line 73 - should Figure 1f have a control that is meant to have poor binding affinity? Bmax is not defined

Page 4 (also Page 5 lines 93-96) – I am not sure that I fully understand the comment on CD52 expression in TCR1 containing cells. It seems that if you are giving TCR1 to a whole Treg fraction using a lentivirus, then the expression of CD52 in a CD4+ cell population responding to the Sm antigen in vitro is not necessarily reflective of expression in a virally transduced population. Is the thought that TCRs associated with Treg have stronger binding affinities for self-antigens, and so the fact that TCR1 may naturally be present in a more regulatory cell type (high CD52+) is meant to support the finding that it has high affinity binding to Sm peptides?

Page 5 – line 88 - For Figure 1g, it would be easier for the reader if you added the label of peptide present (top) and absent (bottom) to the figure in addition to the figure legend. This could be along the left or right side to indicate it applies to the row of images. Also, please indicate what the arrows are pointing to (one assumes the immune synapse, but this should be specified). What is the timing of the interaction (e.g. are they in co-culture for an hour or a day)?

Page 5 – lines 98-99 – please mention the general design of your lentivirus, as the methods are at the end. One infers on the next page that it contains a GFP reporter but this should be stated in text. Also, what kind of promoter is used for expression? Consider including a diagram of the engineered receptor construct in the Figures.

Page 6 – lines 109-113 – refers to extended data Figure 6 – The colors in this diagram are green and blue, but the figure caption refers to unshaded orange. Also, if the polyclonal Tregs are mock-transduced, it would be ideal to use that as the label, because otherwise,

one wonders if the lower GARP expression in Sm-TCR Treg is from the insult of the viral transduction process. In text, it also mentions that the mock-transduced Treg have lower CD69 and GARP expression, but if I am interpreting Extended Data Figure 6 correctly, it looks like GARP expression is actually higher than in Sm-TCR Treg that did not get exposed to Sm antigen (or only low doses)

Page 6 lines 119-120 refer to Extended Data Figure 7; it is a bit unconventional to show cell proliferation studies with a MFI; could you display as the histogram with +/- CTV fractions or give a proliferation index? Please indicate the Treg:Tcon ratio used in the Figure legend. Also please indicate on the figure that the Tcon are TCR1 transduced. Since you leave their native TCR in place, I did wonder if you get any allo-reactivity in these studies. What is the proliferation when Sm antigen is not provided? Lastly, in text, it mentions that a dose-dependent response is shown (line 119), but I do not see a dose titration of Treg

Page 7 line 130, refers to Extended Data Figure 8. This figure shows the Treg are split into half getting transduced with TCR1 and half that are not (presumably mock?). Since the transduction efficiency is only ~20% and you are not sorting, is it correct to conclude that in the in vitro and in vivo functional studies you are comparing mock transduced to 20% TCR1 transduced Treg? Along, these lines, was there any "plasticity" (conversion to IL-17 producers) of transduced and non-transduced Treg cells?

Page 7 line 145 refers to the humanized lupus model. Has this been published before? Given there are many well established models of lupus, it might be helpful to provide more validation data for a new model. I recognize the need for this model for what you are testing, but lupus is complicated, and so additional data could help make the model more convincing.

Page 8 – line 167-168 – more discussion is required for the limitations of the current study and the many additional steps required to advance to Phase I clinical trial. See Overview Comments.

Page 12 – line 261 'CD4+' is repeated

Page 24 – please add mouse age range at Day 0

Reviewer #2 (Remarks to the Author):

This manuscript describes the development of Smith-specific regulatory T cells. The authors identify smith-specific T cells after performing binding assays to identify peptides of smith antigens that bind HLA DR15. They then use these peptides to stimulate PBMC and isolate CD4 T cells that proliferate in response to these antigens. The TCR are sequenced then re-expressed on Jurkat cells to demonstrate TCR specificity, affinity and their ability to promote T cell activation. Finally the investigators express an HLA DR15 restricted sm-specific TCR in nTreg and perform assays to assess suppression in vitro and in a novel NSG model of lupus nephritis.

The manuscript is quite brief, and has limited figures with the majority of information included in the methods and in the supplemental data sections. The identification of HLA DR15 binding peptides is a strength, but there is very limited discussion of how peptide binding

was determined in the main body of the text or in the methods. Further it is unclear why so much information is shared related to protein structure of the HLA-peptide as this does not seem to be the main point of the manuscript.

Selection of the TCR is unusual as the TCR was derived from a healthy male, whereas it may have been more useful to isolate a TCR from a patient with SLE nephritis. TCR clonal expansion is discussed but the sequences were derived from in vitro expanded cells and thus it does not represent in vivo clonal expansion, similarly RNAseq characterization of these cell populations may be skewed by in vitro culture.

The authors then have a good bit of data related to the TCR that they select to demonstrate structure and antigen specificity- again this data does not add much as there are no comparisons across TCR for example or discussion of how these data would influence selection of a TCR for Treg expression.

The Treg assays are non-standard and need to be much more thoroughly described with more information than shared. Treg to Teff ratio after 4 days in coculture is not a typical way to assess suppression and can be skewed by cell death. The authors should consider more well accepted approaches to assessing suppressive function of antigen specific Tregs.

The investigators develop a "novel" NSG model to test the human Treg. This model is only characterized with respect to nephritis- which is actually driven by CD8 T cells and nephron necrosis. This does not seem to be similar to LN. It would be important if the authors plan to publish a new model of LN that they more extensively characterize it. For example do these animals develop GVHD-like symptoms with weight loss?

Finally, the extended Data Fig. 1 outlining a pathway to create antigen specific Treg for autoimmunity is naïve, not taking into account the breadth of testing that would be needed to select a TCR (avoiding cross reactivity for pathogens for example), comparing TCR with each other to identify optimal TCR characteristics.

Reviewer #3 (Remarks to the Author):

This article describes generation and activity of antigen-specific regulatory T cells (Tregs) which are potential therapeutic candidates for systemic lupus erythematosus (SLE).

The authors evaluated potential of Sm-specific Tregs (Sm-Tregs) to suppress disease. Using biophysical affinity binding assays they characterized immunodominant Sm T cell epitope, then identified high-affinity Sm-specific T cell receptors (TCRs) using high-throughput single-cell sequencing. Using lentiviral vectors, then transduced a lead Sm-specific TCR onto Tregs derived from patients with SLE who were anti-Sm and HLA-DR15 positive and analysed the activity of the engineered Tregs in a humanized mouse model of lupus nephritis.

I am worried about MOI used for transduction of Tregs. Was it 25 (or 30 as in ex Fig 5)? The cells were mixed with the vector for a week without any wash? What was the percentage of dead cells in the cultures? Only 20-30% as ex Fig 5?

Pulsing APCs with the peptide. How the authors are sure the peptide was not processed intracellularly before presentation with DR15?

The evaluation of the activity of Tregs after 44 days in the culture? This is relatively long culture for Tregs. For example, refer your work to Cell Transplant. 2011;20(11-12):1747-58.

What was the level of foxp3 methylation, the expression of CD45RA in cultured Tregs after 44 days? It looks from Fig.5 that the majority of the cells are FoxP3low which suggest exTreg phenotype. How stable was the expression of transduced TCR throughout the culture? Do you have any data on TCR expression with time of the culture? What was the yield of proliferating Tregs in the cultures?

How you can distinguish between antigen-specific and allospecific stimulation when peptide-pulsed B-LCLs HLA-mismatched to Tregs and Tconv or Jurkat are used in the suppression tests?

I am wonder how the suppression test would look like if GFP- Tconv from Figure 11b would be taken into account?

Technical:

The presentation of results as mean \pm SEM is odd. It should be rather SD or quartiles and not SEM.

In Fig. 11a , mock-transduced Tregs are presented as TCR versus FCS. I believe it should be TCR versus GFP to ensure readers they are GFPnegative.

In general, supplementary figures are of bad quality. The size of fonts is too small in many cases or the text is cut (See Fig 11b SSc-H versus ?). The gates used in flow analysis are not clearly displayed.

REVIEWER COMMENTS

Reviewer #1 (Remarks to the Author):

*Cheong et al. report the design of Sm-TCR specific Tregs for HLA-DRB1:15 positive individuals and demonstrate that they **can suppress** T cell activation in vitro and kidney damage in vivo in a humanized model of lupus nephritis. They first identify 15mer peptide fragments that span Sm proteins to identify those that bind with highest binding % and affinity to HLA-DRB1:15. They then identify the crystal structure of HLA-DRB1:15 interacting with the top peptide (SmB/B'58-72). They use TCR sequencing and co-culture studies to identify a TCR clonotype that recognizes this complex with high affinity (TCR1), and then they incorporate the TCR alpha and beta chains for this TCR into a lentivirus that they can then transduce conventional and regulatory T cells with to conduct functional studies. The in vivo model used is an MHC null immunodeficient mouse that has been provided with patient PBMCs and later mo-DCs with Sm antigen provided. Mice were treated at week 3 with autologous Sm-TCR specific Tregs and readouts were at week 3 and week 8. Strengths of this paper include the technological sophistication and the logical step-wise approach to designing antigen specific Tregs and testing them in vivo. It is also logistically challenging to use patient cells, and this is an elegant addition to the paper.*

However, I was surprised by how many figures were put in the Supplemental and wonder about bolstering the main text with additional data. I also think the Methods could be condensed - some descriptions could be removed for basic lab protocols or repeated steps (e.g. incubator conditions, mentioning how blood is obtained for each experiment). By shortening the Methods, this could allow greater word count availability to frame the work within the context of the field.

We thank the reviewer for their appraisal of the manuscript. The paper was originally formatted for Nature Medicine as a Brief Communication. Since the manuscript has been reviewed now in Nature Communications, we have updated the manuscript according to the new guidelines. The majority of the data is now contained within the main text and 10 main figures and much less in the supplemental data. We have also updated the methods to be more concise by removing ~2000 words.

Figure 2: I wonder if ab responses were measured and whether anti-Sm was specifically suppressed. This point should pin-point the claimed specificity.

The human cells used in the humanised mouse model were from anti-Sm positive lupus donors. However, in the mouse model it was not possible to measure antibody responses. This is because the NSG mouse strain does not support the long-term engraftment of human B cells. The model does support the long-term engraftment of CD4+ and CD8+ T cells as well as Tregs. See Brehm, FASEB, 2019, 0.1096/fj.201800636R. Given the lack of humoral immunity in this mouse model it is expected the injury is driven by T cell mediated autoimmunity. Considerable literature emphasises the important role of cell mediated immunity in glomerulonephritis, including in lupus nephritis, but we acknowledge this aspect of the mouse model and have specifically addressed this point with an extended discussion (Page 10, line 225-228):

“The NSG mice strain used does not support the long-term engraftment of B cells³¹ and as a result antibody responses were not studied. It is therefore hypothesised the kidney injury observed from anti-Sm⁺ lupus nephritis patient donor cells is driven by T cell mediated autoimmunity.”

Importantly, patient derived anti-Sm autoimmunity was shown to be specifically suppressed in *in vitro* assays whereby patient cells were stimulated with the dominant Sm T cell epitope and the secretion of pro-inflammatory cytokines reduced significantly when Sm-TCR Tregs were added to culture (new Figure 9).

Many other parameters of kidney injury are not mentioned or discussed. For example, cell infiltration in the glomeruli, mesangial cell proliferation, I assume there were no crescents, Tubular epithelial damage, interstitial inflammation and so on.

We have performed additional histological assessments of periodic acid Schiff-stained kidney sections from this model to include glomerular hypercellularity, mesangial cell proliferation and tubulointerstitial injury (see updated Figure 10 d,e,f). Analyses of these parameters were concordant with the other parameters in that the administration of Sm-Tregs resulted in reduced kidney injury compared to administration of polyclonal Tregs or no Tregs. We have also included an extra control group of donors who were healthy (i.e. without lupus nephritis) primarily to assess whether any graft-versus-host disease was present. The reviewer is correct to assume there were no crescents found by histological assessment. We have updated the results and discussion to include (Page 11, line 244-251):

“The extent of glomerular hypercellularity was significantly lower in mice administered Sm-Tregs (25.8 cells/glomerular cross section(gcs)) compared to those administered polyclonal Tregs (37.8 cells/gcs) or no Tregs (39.8 cells/gcs) (Fig 10d). Mesangial cell proliferation was significantly reduced in Sm-Treg administered mice (14% glomeruli affected) compared to polyclonal Tregs (59%) or no Tregs (62%) (Fig 10e). Tubulointerstitial injury was also significantly reduced in mice receiving Sm Tregs (10% area affected) compared to polyclonal Tregs (50%) or no Tregs (50%) (Fig. 10f). Glomerular crescents were not observed in any sample.”

Some questions that I have, for example, include whether the authors think that the Tregs are predominantly acting through contact dependent vs. independent manners. I think it is clear that the Treg cells are activated by a specific antigen – this is by design. However, are they recognizing the Sm peptide and acting by bystander suppression, are they blocking Tcon from binding to the MHC-peptide (especially TCRI+ Tcon), are they killing activated T cells, and/or are they otherwise exerting their regulatory effect on Tcon/APCs?

This is a good point to raise and we believe the Tregs are working in a variety of ways. The mechanisms through which Sm Tregs can act are now diagrammatically summarised in Fig. 1. Firstly, the engagement of the Sm-specific TCR by peptide-MHC acts to activate the Treg, which in turns enhances its suppressor functions through both antigen-specific and bystander effects. We have added data to the manuscript to support this by addressing the following mechanisms:

- With regard to bystander suppression, we have included in Extended Data Fig 1b results showing that the Sm Tregs more potently suppress the polyclonal Tconv cell proliferation than polyclonal Tregs. This is likely due to the Sm Tregs being more activated upon presentation of Sm peptide by MHC to Sm TCR. This could have an effect on restoring immune tolerance to other autoantigens involved in lupus nephritis. We have included the following text in the results and discussion, as follows (Page 8, line 178-182):
- “Sm Tregs were also able to show a superior dose-dependent suppressive bystander effect on polyclonal Tconv proliferation compared to mock-Tregs (Extended Data Fig. 1b). This

shows that in the presence of SmB/B'58-72 autoantigen, the Sm Tregs can better suppress Tconvs of other specificities, an effect which contributes to restoring immune tolerance.”

- With regard to tolerization of APCs, we have added data that show B cell tolerization. When a DR15⁺ B cell line (B-lymphocytoblastoid, B-LCL) is cultured with Sm or mock Tregs, heightened tolerization of the B-LCL is observed in the presence of the Sm Tregs compared to mock Tregs, as measured by reduction in surface expression of CD80, CD86 and HLA-DR. We have included these data in Extended Data Fig. 2 and in the main body as follows (Page 9, line 184-192),

“Another mechanism Tregs utilize to suppress autoimmunity is through the tolerization of antigen presenting cells (APCs) such as mature DCs and B cells^{25,26}. The tolerizing effects of Tregs acts to reduce the capacity of APCs to present autoantigens to pathogenic T cells through contact-dependent and contact-independent mechanisms. Tregs can also suppress B cells by reducing their ability to secrete autoantibodies and present autoantigens^{27,28}. Compared to mock Tregs, Sm Tregs were able to better tolerize CD19⁺ B-LCL cell lines as shown by a significant reduction in the expression of HLA-DR, meaning reduced capacity to present autoantigens by HLA as well as a significant reduction in expression of co-stimulatory ligands CD80 and CD86 (Extended Data Fig. 2).”

- With regard to contact dependent vs. independent manners, cytokines were measured from LN patient PBMCs. An increase in the anti-inflammatory cytokine IL-10 and decreases in pro-inflammatory cytokines IL17A and IFN-gamma was observed after the addition of Sm Tregs compared to polyclonal Tregs or no Tregs (see Fig 9). Although not a trans-well experiment, this suggests that contact-independent mechanisms such as by cytokine signalling are at least one way in which Sm Tregs can exert their effects on autoimmunity.

*This matters because if framed as being antigen specific, the question arises as to **how** they would behave in human lupus in which there are many auto-antigens present. The in vivo model is purely based on Sm antigen and does not test this. Meanwhile, if Sm peptides are used to activate Treg that act by bystander suppression, this might cover the more complex autoimmune situation in patients, but then it would be a broader type of immune suppression (not antigen specific). Either way would potentially be useful, so this is not a weakness but rather something that could be discussed. This approach implies that such Sm-specific Tregs will be used in Sm positive lupus patients under the assumption that these abs account for or are involved in disease pathogenesis. And a question to discuss, is how they propose to choose autoantigens beyond Sm.*

We thank the reviewer for highlighting this important point. The ability to induce clinically significant nephritis via exposure to Sm peptide and human Sm-reactive patient PBMCs strongly suggests a causal role for anti-Sm autoreactivity in LN. We have shown that Sm-Tregs act in antigen-specific ways. However, while this does not prove that no other autoreactivities are implicated in LN, the effect of Sm-Tregs do not stop at the antigen-specific immune suppression, but rather the heightened active state of Sm Tregs in the context of LN induced with Sm antigen present can also give rise to bystander effects. Such bystander effects could have an immune tolerizing effect on other autoantigens. We have included an extra figure on the suppressive effect of Sm Tregs on bystander Tconvs (Extended Data Fig. 1b) and have updated the discussion to include (Page 8, line 178-182):

“Sm Tregs were also able to show a superior dose-dependent suppressive bystander effect on polyclonal Tconv proliferation compared to mock-Tregs (Extended Data Fig. 1b). This

shows that in the presence of SmB/B'₅₈₋₇₂ autoantigen, the Sm Tregs can better suppress Tconvs of other specificities, an effect which contributes to restoring immune tolerance.”

With regard to how we propose to choose autoantigens beyond Sm in future work, we highlight a development process from autoantigen discovery, TCR discovery, TCR Treg product characterization through to clinical trials and have included a new Figure 2 to illustrate this process. This process could be used to look at autoantigens implicated in SLE beyond Sm, such as Ro and La, as well as other HLA-restrictions in SLE. In addition, the process can potentially be used to target other HLA-linked autoimmune diseases. In the concluding remarks we have added (Page 12, line 269-272):

“Using LN as a proof of principle, we have developed a platform to establish TCR Treg cell based therapy that has the potential to be extended into other autoantigens and HLA-types implicated in manifestations of SLE as well as in other HLA-linked autoimmune diseases.”

Not directly relevant to this paper but worth mentioning for translational considerations is the short half-life of Treg in vivo and the IL-2 deficient state of patients with SLE that may lead to an even shorter half-life. I noticed the IL-2 dose used in culture is 1000IU/mL for culture from healthy buffy coat and 300IU/mL for those isolated from lupus patients. Is the reason for this related to the second scenario not being an expansion phase?

Would the use of lower dose go against the fact that IL-2 production is low in SLE, and, more importantly, that T cells from patients with SLE do not generate sufficient amounts of pSTAT5 in response to IL-2? Given the flurry of efforts to treat patients with lupus with low dose IL-2, should Ag-specific Tregs be used in conjunction with IL-2? – a discussion point.

We thank the reviewer for their comments regarding IL-2. It remains unknown whether IL-2 adjunct therapy would be beneficial in Sm Treg therapy for LN. Given the IL-2 deficient state of SLE patients, it is a worthwhile consideration, but we note that randomised trials of IL-2 administration in SLE have been disappointing (He J et al, Ann Rheum Dis, 2020; Humrich JY et al, Ann Rheum Dis, 2022). We have not directly addressed whether IL-2 supplementation is beneficial in our humanised mouse model of LN. We expect that the IL-2 needed for Treg persistence is derived from the adoptively transferred patient Tconvs. Other Treg therapy studies such as Bluestone’s polyclonal Treg trial in T1D showed that adoptively transferred Tregs are not short-lived and can remain at up to 25% of the peak level in the circulation at 1 year after transfer without IL-2 adjunct therapy after transfer (Bluestone JA, Sci Transl Med, 2015). This paper also showed that high dose IL-2 (300IU/mL) during the in vitro expansion step before adoptive transfer, caused the autologous donor Tregs to express enhanced phosphorylation of pSTAT5. Given SLE and other autoimmune disease patients show reduced phosphorylation of STAT5 as a marker of Treg dysfunction, it is encouraging evidence that the in vitro expansion conditions we used can restore Treg function in this regard.

To address these points, we have added to the discussion (Page 12, line 254-265):

“In the Sm Treg in vivo mouse model, we did not administer any IL-2 adjunct therapy, as previously reported clinical trials of Tregs in autoimmune disease did not supplement IL-2 after transfer and Tregs remained in the patients for up to 1 year after transfer suggesting IL-2 supplementation is not necessary⁴. Randomised trials of low-dose IL-2 to restore Treg function in SLE have been disappointing³³ despite initial encouraging results in open label studies which aimed to address Treg and IL-2 deficiency in SLE patients³⁴. A previous report of Treg therapy in humans showed the Treg in vitro expansion with high-dose IL-2 prior to transfer, a similar method to ours, can restore donor Treg function in T1D by increasing phosphorylation of signal transducer and activator of transcription 5 (pSTAT5),

a marker of IL-2-dependent Treg function⁴. Given the in vitro supplementation of IL-2 was sufficient to restore Treg function and promote long-term engraftment we deemed IL-2 adjunct therapy unnecessary for Sm Tregs.”

In relation to the concentration of IL-2 used for in vitro Treg expansion, IL-2 is added every 2 days in order to achieve the required Treg numbers and stable phenotype for therapy. We have used both 300 and 1000 IU/mL IL-2 for this in vitro Treg expansion and based on previously published literature both concentrations are considered high-dose. Comparatively, Tconvs only require approximately 40IU/mL IL-2. Both 300 and 1000IU/mL IL-2 gave comparable Treg fold increase and phenotype.

Text Comments

Abstract line 24 – perhaps it is meant to say “our lead”?

Changed to “our lead”

Page 3 line 37 – ‘high mortality’ is a bit vague; it may be more accurate to say that ‘LN is an important contributor to disease related morbidity and mortality’

Changed to “Lupus nephritis (LN), a severe manifestation of SLE, is an important contributor to disease related morbidity and mortality”

Page 3 line 41 – is there a word missing after ‘Sm antigen-specific’?

Changed to Sm antigen-specific Treg.

Page 3 line 51 – could include parenthetical statement regarding the meaning of ‘stability index’, either in text or the figure legend; I see how the calculation is made in the Methods, but perhaps a brief descriptive statement in the main text could help the reader understand the conclusion

We have included a definition of ‘stability index’ in the main text, “calculated as the peptide-HLA half-life multiplied by the binding assay Reveal score.” We also extended the calculation steps in the methods.

Page 4 – line 73 - should Figure 1f have a control that is meant to have poor binding affinity? Bmax is not defined

We thank the reviewer for their suggestions to improve Fig. 1f, now Fig. 6b. We have included a negative control in the graph being the same cells stained with negative control dextramer, DR15 human CLIP₁₀₃₋₁₁₇, which is not expected to bind Sm TCR (dotted grey curve). We have defined Bmax in the figure legend as “the maximum specific binding in units of Dextramer PE mean fluorescence intensity (Bmax).”

Page 4 (also Page 5 lines 93-96) – I am not sure that I fully understand the comment on CD52 expression in TCR1 containing cells. It seems that if you are giving TCR1 to a whole Treg fraction using a lentivirus, then the expression of CD52 in a CD4+ cell population responding to the Sm antigen in vitro is not necessarily reflective of expression in a virally transduced population. Is the thought that TCRs associated with Treg have stronger binding affinities for

self-antigens, and so the fact that TCR1 may naturally be present in a more regulatory cell type (high CD52+) is meant to support the finding that it has high affinity binding to Sm peptides?

Yes, the reviewer is correct to assume the CD52 expressing Treg cells that also express TCR1 from the scRNAseq reinforces that TCR1 is specific for a self-antigen since it is understood that in healthy humans the majority of Tregs that exit the thymus are on a Treg background to maintain immune homeostasis. For clarity, we have added an extra discussion sentences (Page 5, line 99-106):

“Interestingly, the significant upregulation of CD52 in cells expressing TCR1 suggests that TCR1 is associated with a type of nonconventional Treg based on previous research in type 1 diabetes (T1D) showing that CD4⁺CD52^{hi} cells possess regulatory properties¹⁴. Other significantly upregulated genes on the TCR1⁺ subset were also associated with Tregs, such as *LAIR2*, expressed on Tregs in other scRNASeq studies, and *IL9R* is associated with Treg suppressive potency^{15,16}. The Treg phenotype of the TCR1 expressing T cell subset reinforces that the specificity of TCR1 is for an autoantigen since affinity for self-antigens promotes the selection of Tregs¹⁷.”

Page 5 – line 88 - For Figure 1g, it would be easier for the reader if you added the label of peptide present (top) and absent (bottom) to the figure in addition to the figure legend. This could be along the left or right side to indicate it applies to the row of images. Also, please indicate what the arrows are pointing to (one assumes the immune synapse, but this should be specified). What is the timing of the interaction (e.g. are they in co-culture for an hour or a day)?

We thank the reviewer for their suggestions to improving the imaging flow cytometry figures (now Fig. 6). We have included labels for with and without peptide in the figure and figure legend. We have also updated the figure legend to include definition of the arrow and the time of interaction, as follows,

“Image flow cytometry of the interaction of c, TCR1, e, TCR2, g, TCR3 on J76 Jurkats incubated for 2hr with HLA-DR15 B-LCLs in the presence (above) and absence (below) of SmB/B’₅₈₋₇₂. Arrows point to the mature immune synapse.”

Page 5 – lines 98-99 – please mention the general design of your lentivirus, as the methods are at the end. One infers on the next page that it contains a GFP reporter but this should be stated in text. Also, what kind of promoter is used for expression? Consider including a diagram of the engineered receptor construct in the Figures.

We have included the general design of the lentivirus in the results/discussion section and have added a diagram of the lentiviral construct in Fig 6a.

“Lentiviral vectors encoding TCR1-3 were created with an EF1alpha promoter controlling the expression of a tricistronic vector encoding the TCR beta chain, TCR alpha chain and eGFP reporter interspaced with P2A and T2A ribosomal skipping sequences (Fig. 6a). The lentiviral vectors were used to transduce J76 Jurkat T cell line lacking endogenous TCR and primary human Tregs.”

Page 6 – lines 109-113 – refers to extended data Figure 6 – The colors in this diagram are green and blue, but the figure caption refers to unshaded orange.

This has been corrected and now forms part of Fig. 8a-b.

Also, if the polyclonal Tregs are mock-transduced, it would be ideal to use that as the label, because otherwise, one wonders if the lower GARP expression in Sm-TCR Treg is from the insult of the viral transduction process.

Polyclonal Tregs have been renamed as polyclonal mock-transduced.

In text, it also mentions that the mock-transduced Treg have lower CD69 and GARP expression, but if I am interpreting Extended Data Figure 6 correctly, it looks like GARP expression is actually higher than in Sm-TCR Treg that did not get exposed to Sm antigen (or only low doses)

Yes, looking at GARP on Sm Tregs does appear lower with no or little Sm antigen compared to mock Tregs. We have not established why there is a difference here. The main difference we wish to highlight is that there is a significantly increased CD69 and GARP expression in the Sm Tregs compared to mock Tregs and that this expression of activation markers is Sm antigen dose-dependent. We have also added to the manuscript activation assays using lupus patient-derived Tregs and observed similar upregulation of activation markers CD69, LAP and GARP on Sm Tregs compared to untransduced/mock Tregs (Extended Data Fig. 3).

*Page 6 lines 119-120 refer to Extended Data Figure 7; it is a bit unconventional to show cell proliferation studies with a MFI; could you display as the histogram with +/- CTV fractions **or give a proliferation index**? Please indicate the Treg:Tcon ratio used in the Figure legend. Also please indicate on the figure that the Tcon are TCR1 transduced. Since you leave their native TCR in place, I did wonder if you get any allo-reactivity in these studies. What is the proliferation when Sm antigen is not provided? Lastly, in text, it mentions that a dose-dependent response is shown (line 119), but I do not see a dose titration of Treg.*

The suppression assay has been updated to refer to ‘suppression’ in terms of ‘% suppression’ defined as the increase in CellTrace Violet MFI (decrease in proliferation) from Sm Tconvs compared to the baseline proliferation of Sm Tconvs without Tregs (un-suppressed). Ratios of Treg:Tconv have been used as well as updating the Treg dose response. The main text now addresses that alloreactivity cannot be assessed in this assay, “the extent of allo-specific stimulation was unable to be assessed in this assay”. This is now in new Fig. 8c.

Page 7 line 130, refers to Extended Data Figure 8. This figure shows the Treg are split into half getting transduced with TCR1 and half that are not (presumably mock?). Since the transduction efficiency is only ~20% and you are not sorting, is it correct to conclude that in the in vitro and in vivo functional studies you are comparing mock transduced to 20% TCR1 transduced Treg?

Yes, when we refer to TCR1 transduced Tregs we are referring to a population of 20% TCR1 transduced Tregs. We have used this % because it correlates with a vector copy number that is less than 5, which is the current criteria for FDA approval for gene engineered cell based therapies.

Along, these lines, was there any “plasticity” (conversion to IL-17 producers) of transduced and non-transduced Treg cells?

We performed intracellular cytokine staining to look at the frequency of IL-17A and interferon gamma on 10 day in vitro expanded Sm Tregs and the Sm Tregs showed low expression of IL-17A and interferon gamma. These data have been added in the revised manuscript by including extra figures in Figure 7f-g and in the main text (Page 7, line 150-158):

“Concordant with a stable Treg phenotype, Sm Tregs showed low expression of pro-inflammatory cytokines IL-17A and IFN- γ measured by intracellular flow cytometry (Fig. 7f-g). Stimulated Tregs at 10-days in vitro expansion showed 5.4-fold less IL-17A expression (mean 1.1% IL-17A⁺) compared to stimulated Tconvs (mean 6.1% IL-17A⁺). Stimulated Sm Treg IFN- γ expression was similarly low at 4.6% IFN- γ ⁺ compared to 26.7% IFN- γ ⁺ in stimulated Tconvs. Both IL-17A and IFN- γ expression was lower in Sm Tregs compared to un-transduced or mock Tregs. Sm Tregs are therefore able to maintain their suppressor phenotype as there is minimal conversion to pro-inflammatory Th17 cells in vitro.”

Furthermore, we performed analyses to determine the stability of Sm Tregs by assessing methylation of the Treg cell specific demethylated region (TDSR) of the FOXP3 locus. This showed a consistently de-methylated phenotype of in vitro expanded Sm Tregs which is consistent with a stable Treg phenotype. We have included an extra heatmap in Fig 7e to show this data and have included in the main body (Page 7, line 143-150),

“Sm Treg regulatory phenotype stability was further confirmed by methylation analysis of the Treg-cell specific de-methylated region (TDSR) of the FOXP3 locus (Fig. 7e). After 3 weeks of in vitro expansion, Sm Tregs maintained a mean TDSR methylation of 52.8%, consistent with ex vivo Tregs (44.5%) and better than the 3-week-expanded mock Tregs at 70.73% methylated. Conversely ex vivo Tconvs showed a mean TDSR methylation of 87.43% and 3-week in vitro expanded Tconvs 90.33% methylation indicating that the Sm Tregs showed consistent TDSR de-methylation long-term, a hallmark of a stable Treg phenotype.”

Page 7 line 145 refers to the humanized lupus model. Has this been published before? Given there are many well established models of lupus, it might be helpful to provide more validation data for a new model. I recognize the need for this model for what you are testing, but lupus is complicated, and so additional data could help make the model more convincing.

This is a novel humanised lupus model. We have extended the mouse model data to include extra validation data. These include extra histological assessments of kidney injury (glomerular hypercellularity, mesangial cell proliferation and tubulointerstitial injury), as noted above. These measurements were consistent with the previously included kidney injury data of glomerular necrosis and proteinuria. These are included in Fig. 10d-f.

As an extra control group, healthy human donor cells were used to ensure the mice do not spontaneously develop disease when human cells are adoptively transferred and therefore that the kidney injury observed is due to the pathogenic immune cells from the LN patients. Previously published results show these mice do not spontaneously develop graft-versus-host disease when healthy human cells are adoptively transferred. As expected, healthy human donor cells in the model produced no increased proteinuria at week 8 (updated Fig. 10b). The healthy human cell control mice also did not develop kidney injury (updated Fig. c-g).

Page 8 – line 167-168 – more discussion is required for the limitations of the current study and the many additional steps required to advance to Phase I clinical trial. See Overview Comments.

We have included a new Fig. 2 to visually show the steps required to advance to a phase I clinical trial. We have included extra detail in the conclusions and future directions section to address the reviewer's concerns (Page 12, line 266-277):

“In conclusion, we have successfully identified HLA-DR15-restricted, Sm-specific, high-affinity TCRs using a novel method of physical binding assays, DC and PBMC co-cultures, and high throughput single-cell sequencing. We have also shown that these Sm-specific TCRs, once transduced onto primary patient Tregs, have conferred Sm-specificity and the ability to functionally suppress Sm-autoreactivity both in vitro and in vivo. Using LN as a proof of principle, we have developed a platform to establish TCR Treg cell based therapy that has the potential to be extended into other autoantigens and HLA-types implicated in manifestations of SLE as well as in other HLA-linked autoimmune diseases. We aim to further characterize Sm Tregs in our novel humanized LN mouse model by tracking Sm Tregs long-term in vivo and assessing their persistence and phenotype. The Sm Treg production method is being developed into a GMP-ready, closed-system method of manufacture, which can then be used as a cell-based therapeutic in a phase I clinical trial for patients with LN.”

Page 12 – line 261 ‘CD4+’ is repeated

Repeated word removed.

Page 24 – please add mouse age range at Day 0

Age range has been included in the methods as ‘8-10-week-old’.

Reviewer #2 (Remarks to the Author):

This manuscript describes the development of Smith-specific regulatory T cells. The authors identify smith- specific T cells after performing binding assays to identify peptides of smith antigens that bind HLA DR15. They then use these peptides to stimulate PBMC and isolate CD4 T cells that proliferate in response to these antigens. The TCR are sequenced then re-expressed on Jurkat cells to demonstrate TCR specificity, affinity and their ability to promote T cell activation. Finally the investigators express an HLA DR15 restricted sm-specific TCR in nTreg and perform assays to assess suppression in vitro and in a novel NSG model of lupus nephritis.

The manuscript is quite brief, and has limited figures with the majority of information included in the methods and in the supplemental data sections.

We thank the reviewer for their appraisal of the manuscript. The paper was originally formatted for Nature Medicine as a Brief Communication. Since the manuscript has been reviewed now in Nature Communications, we have updated the manuscript according to the new guidelines. The majority of the data is now contained within the main text and 10 main figures and much less in the extended data. We have also updated the methods to be more concise by removing ~2000 words.

The identification of HLA DR15 binding peptides is a strength, but there is very limited discussion of how peptide binding was determined in the main body of the text or in the methods.

We acknowledge the limited description of the peptide binding assay in the originally submitted brief manuscript and have updated the main body and methods to adequately address how Sm peptides were screened for HLA-DR15 binding and stability.

Main text added (Page 3, line 55-60):

“Of all the Sm peptides tested, SmB/B’₅₈₋₇₂ had the highest binding to HLA-DR15 as measured by the % binding relative to a unique positive control peptide bound to HLA-DR15. SmB/B’₅₈₋₇₂ also showed the greatest stability for HLA-DR15 as measured by the Stability Index, which measures the stability of each peptide-HLA-DR15 complex as a value normalised to the half-life of the peptide-HLA and the binding score (see methods).”

Methods added (Page 13, line 281-297):

“A biophysical affinity binding assay, MHC Class II Reveal (Proimmune), was used to determine HLA-DRB1*15:01 (HLA-DR15)-restricted T cell epitopes 35. A total of 145 15-mer peptides offset by three amino acids from SmB/B’ (GenBank Accession: AAA36578.1, AAA36579.1), SmD1 (GenBank Accession AAH72427.1) and SmD3 (GenBank Accession AAH03150.1) were screened against HLA-DR15 at 37°C. If the peptide bound to HLA DR15, a specific-monoclonal antibody bound and a fluorescent signal was released. Fluorescent signal measurements of peptide-HLA binding were taken at 0h and 24hr and given as Reveal scores, defined as peptide-HLA binding relative to a positive control peptide, which is a known T-cell epitope for HLA-DR15. The Reveal scores were used derive a half-life approximation based on a one-phase dissociation equation. Specifically, the equation to fit data to generate off-rates was $Y = A1 * EXP(-A2 * X)$ where Y = the Reveal Score at 24hr, A1 = the Y[0] also known as the Reveal Score at 0hr and A2 = k also known as the rate constant. The rate constant, k, was defined as $k = -((\ln(Y/Y[0]))/X)$, where x = time in hours. Then the rate constant (k) was used to calculate the on- and off-rate half-life values (t1/2) according to the equation $t1/2 = \ln2/k$. For analysis, a stability index score was used as a normalised measure of peptide-MHC binding by the following equation; Stability Index = $t1/2 * Y[0]/100$. The Stability Index was used to weight the value to minimize over-representation of weakly binding peptides.”

Further it is unclear why so much information is shared related to protein structure of the HLA-peptide as this does not seem to be the main point of the manuscript.

The protein crystal structure of HLA-peptide offers a high-resolution view of the antigen-specific interaction and how the Smith autoantigen can present to T cells. Very few crystal structures of autoantigen-HLA have been solved and as such any knowledge of these interactions better characterise antigen-specific interactions in autoimmunity in general, as well as defining the exact nature of antigen presentation driving lupus nephritis. To emphasise the importance of the protein crystal structure we have updated the manuscript to include (Page 4, line 70-79),

“The protein crystal structure of SmB/B’₅₈₋₇₂ in complex with HLA-DR15 proves SmB/B’₅₈₋₇₂ binds HLA-DR15, defines its interaction and confirms the nature of the structure is indeed an antigen-specific interaction that binds in a conventional manner. This novel structure is one of the only three peptide-HLA structures reported to date for human self-antigens in the context of HLA-DR15. Knowledge of the structural characteristics defining which autoantigenic peptide residues bind to HLA-DR15 will assist with the design and improvement future candidate TCRs for lupus nephritis as well as other HLA-linked disease conditions. This structural analysis will also facilitate future off-target screening as key

structural features of the autoantigen that can be recognised by Sm TCRs have been determined.”

Selection of the TCR is unusual as the TCR was derived from a healthy male, whereas it may have been more useful to isolate a TCR from a patient with SLE nephritis. TCR clonal expansion is discussed but the sequences were derived from in vitro expanded cells and thus it does not represent in vivo clonal expansion, similarly RNAseq characterization of these cell populations may be skewed by in vitro culture. Add to discussion the cell populations may be skewed by in vitro culture but nonetheless it is interesting that TCR1 was found on CD52hi cells when the idea that suppressor like cells escape the thymus expressing self-recognising TCRs.

We acknowledge that this mode of TCR selection is novel but it is a key feature of our approach, based on our published work that healthy people are protected from autoimmune disease by the presence of autoantigen-specific regulatory T cells, that are in turn deficient in individuals with disease. Healthy individuals have a very low frequency of autoantigen specific T cells circulating in the periphery and one of the reasons they do not exhibit autoimmunity is that these T cells are of a Treg phenotype. We show that we are able to expand the autoantigen-specific T cells from healthy donors to identify TCRs that are high-affinity for Smith autoantigen. Previously reported techniques for identifying TCRs from patient PBMCs are of course still valid and we add to the literature to show that autoantigen specific TCRs can be identified using healthy human donors.

We agree the scRNASeq data does not reflect the in vivo precursor T cell phenotype of the cells and have addressed this in the main text by adding (Page 5, line 96-99):

“Although scRNASeq data was generated on in vitro expanded CD4⁺ T cells and as such does not reflect the in vivo expression profile of the precursor cells, it is worthwhile to assess the T cell subtypes giving rise to the dominant TCR subpopulations.”

The authors then have a good bit of data related to the TCR that they select to demonstrate structure and antigen specificity- again this data does not add much as there are no comparisons across TCR for example or discussion of how these data would influence selection of a TCR for Treg expression.

We chose one TCR, TCR1, as our lead TCR for complete characterization. This TCR was compared with other TCRs based on clonotypic abundance from scRNASeq revealing TCR1 as the most abundant of all TCRs sequenced (Fig 5b), based on affinity of TCR for pMHC by comparing TCR1, TCR2 and TCR3 revealing TCR1 as the highest affinity (Fig. 6b) and finally based on immune synapse intensity by comparing TCR1, TCR2 and TCR3 revealing TCR1 as making mature immune synapse between T cell and APC (Fig. 6c-h). Therefore, we believe there are sufficient comparisons across TCRs to allow for selection of TCR1 as our lead TCR. We then further characterised TCR1 based on transduced Treg functional responses in vitro and in vivo.

The biological machinery of TCR expression on Treg is expected to be analogous to the that of other CD4⁺ T cells as there is no difference in the nature of the TCR complex between these T cell subtypes. We therefore were able to transduce TCRs originating from Tconvs onto Tregs and show antigen-specific responses.

*The Treg assays are non-standard and need to be much more thoroughly described with more information than shared. **Treg to Teff ratio after 4 days in coculture** is not a typical way to*

assess suppression and can be skewed by cell death. The authors should consider more well accepted approaches to assessing suppressive function of antigen specific Tregs.

We appreciate the comment about the unconventional reporting of Treg to Teff ratio after 4 days figure and have removed it from the manuscript/figure.

The suppression assay has been updated to refer to ‘suppression’ in terms of ‘% suppression’ defined as the increase in CellTrace Violet MFI (decrease in proliferation) from Sm Tconvs compared to the baseline proliferation of Sm Tconvs without Tregs (un-suppressed). Ratio of Treg:Tconv have been used as well as updating the Treg dose response. This is now new Fig. 8c.

The investigators develop a “novel” NSG model to test the human Treg. This model is only characterized with respect to nephritis- which is actually driven by CD8 T cells and nephron necrosis. This does not seem to be similar to LN. It would be important if the authors plan to publish a new model of LN that they more extensively characterize it. For example, do these animals develop GVHD-like symptoms with weight loss?

We have included an extra control group of healthy donors primarily to assess whether any GVHD was present. Mice receiving healthy donor cells in the same way as Fig. 10a did not show any significant proteinuria at week 8 and histological assessment of their kidneys showed no kidney injury. Furthermore, no signs of weight loss (+/- 1g) were observed in the healthy donor cell control group suggesting GVHD is not a concern for these mice. This is in agreement with published reports of these mice showing no signs of GVHD (Brehm, FASEB, 2019, 0.1096/fj.201800636R).

To more extensively characterise the model, we performed further histological assessments of periodic acid Schiff-stained kidney sections to include glomerular hypercellularity, mesangial cell proliferation and tubulointerstitial injury (see updated Figure 10 d,e,f). Analysis of these extra parameters showed concordance with the other parameters such that the administration of Sm Tregs resulted in less kidney injury compared to polyclonal Tregs or no Tregs.

We have updated the results and discussion to include (Page 11, line 244-253):

“The extent of glomerular hypercellularity was significantly lower in mice administered Sm-Tregs (25.8 cells/glomerular cross section(gcs)) compared to those administered polyclonal Tregs (37.8 cells/gcs) or no Tregs (39.8 cells/gcs) (Fig 10d). Mesangial cell proliferation was significantly reduced in Sm-Treg administered mice (14% glomeruli affected) compared to polyclonal Tregs (59%) or no Tregs (62%) (Fig 10e). Tubulointerstitial injury was also significantly reduced in mice receiving Sm Tregs (10% area affected) compared to polyclonal Tregs (50%) or no Tregs (50%) (Fig. 10f). Glomerular crescents were not observed in any sample. To control for graft-versus-host disease (GVHD), a group of mice that received healthy donor cells and un-transduced Tregs showed no signs of weight loss or renal disease that could be attributed to GVHD (Fig. 10b-g), consistent with previous reports using this NSG mouse strain³¹.”

Finally, the extended Data Fig. 1 outlining a pathway to create antigen specific Treg for autoimmunity is naïve, not taking into account the breadth of testing that would be needed to select a TCR (avoiding cross reactivity for pathogens for example), comparing TCR with each other to identify optimal TCR characteristics.

We fully appreciate the complexity and breadth of testing required to select the best TCR to take the Sm Treg to the clinic. We have sought to address this issue by re-making the pathway figure to be more detailed along each step. We have addressed specifically the TCR cross-reactivity studies in panel 3 of the figure. This pathway is now more prominently highlighted as Fig. 2.

Reviewer #3 (Remarks to the Author):

This article describes generation and activity of antigen-specific regulatory T cells (Tregs) which are potential therapeutic candidates for systemic lupus erythematosus (SLE).

The authors evaluated potential of Sm-specific Tregs (Sm-Tregs) to suppress disease. Using biophysical affinity binding assays they characterized immunodominant Sm T cell epitope, then identified high-affinity Sm-specific T cell receptors (TCRs) using high-throughput single-cell sequencing. Using lentiviral vectors, then transduced a lead Sm-specific TCR onto Tregs derived from patients with SLE who were anti-Sm and HLA-DR15 positive and analysed the activity of the engineered Tregs in a humanized mouse model of lupus nephritis.

*I am worried about **MOI** used for transduction of Tregs. Was it 25 (or 30 as in ex Fig 5)? The cells were mixed with the vector for a week without any wash? What was the percentage of dead cells in the cultures? Only 20-30% as ex Fig 5?*

MOI for TCR transduction of Tregs was 30, which is similar to previously published methods for Treg transduction (doi:10.1371/journal.pone.0011726) and found to be optimal for TCR transduction of Tregs. We realised one part of the methods was reported as MOI 25 in error and have updated this to MOI 30.

We also realised the media exchange step the day after transduction was omitted in error from the methods, and have added the detail to the methods: "The next day, media was exchanged with fresh media (as above) and re-incubated."

Pulsing APCs with the peptide. How the authors are sure the peptide was not processed intracellularly before presentation with DR15?

We used mature dendritic cells which have intracellular processing capacity. Therefore, the peptide is able to be processed and presented on DR15.

The evaluation of the activity of Tregs after 44 days in the culture? This is relatively long culture for Tregs. For example, refer your work to Cell Transplant. 2011;20(11-12):1747-58. What was the level of foxp3 methylation, the expression of CD45RA in cultured Tregs after 44 days?

We acknowledge that culturing Tregs for 44 days is a long length of culture and typically we culture Tregs in vitro for 10-14 days. Given there is little relevance to looking at Treg phenotype in vitro at day 44, we have excluded this from the revised manuscript and provided Treg phenotypic data for 10-14 days expansion, which is in line with their use in our in vitro and in vivo assays.

Since the Tregs are stimulated with anti-CD3 and anti-CD28 in order to expand their numbers, they lose expression of CD45RA and this is expected.

We have included new Treg stability data based on lack of expression of IL-17A and IFN- γ and demethylation of the TDSR region. The newly added data shows that Sm Tregs show low expression of IL17A and IFN- γ after 10 days expansion and are predominantly demethylated in the FOXP3 TDSR region after 3 weeks of in vitro expansion (see new Fig. 7e-g).

It looks from Fig.5 that the majority of the cells are FoxP3low which suggest exTerg phenotype. How stable was the expression of transduced TCR throughout the culture? Do you have any data on TCR expression with time of the culture? What was the yield of proliferating Tregs in the cultures?

We have re-assessed FOXP3 and Helios expression in day 10 in vitro expanded Sm Tregs and have shown they are 87% double positive for FOXP3 and Helios. Further analysis mentioned above looking at FOXP3 TDSR, IL17A and IFN- γ all suggest Sm Tregs are of a bona fide Treg phenotype.

Regarding the stability of Sm TCR expression over time in vitro, we have included an extra figure (Extended Data Fig. 1a). We tracked the frequency of Sm TCR in 4 separate in vitro expansion experiments over a course of 10-42 days. We observed no significant drop in transduced Sm TCR expression in the Tregs during this time. This suggests the TCR transgene is able to stably express. The following text was added to the main body,

“Sm TCR1 Tregs maintained stable transgenic expression of TCR1 long-term in vitro as the expression of target TCR1 remained consistently between a mean expression of 16.7% at day 10 and 13.6% at day 42 (Extended Data Fig. 1a)”

How you can distinguish between antigen-specific and allospecific stimulation when peptide-pulsed B-LCLs HLA-mismatched to Tregs and Tconv or Jurkat are used in the suppression tests?

The assays do not exclude allospecific stimulation. We do highlight that in this case any allospecific Treg stimulation would be anti-inflammatory, not pro-inflammatory. This is a limitation of the study and we have addressed it in the main body (Page 8, line 177-178):

“...the extent of allo-specific stimulation was unable to be assessed in this assay.”

I am wonder how the suppression test would look like if GFP- Tconv from Figure 11b would be taken into account?

The GFP- Tconv proliferation is essentially looking at Tconv bystander suppression, which we have now included in Extended Data Fig. 1b showing that the Sm Tregs more potently suppress polyclonal Tconv cell proliferation compared to polyclonal Tregs. This is likely due to the Sm Tregs being more activated upon presentation of pMHC to Sm TCR. This could have an effect on restoring immune tolerance to other autoantigens involved in lupus nephritis. We have included the following text in the results and discussion (Page 8, line 178-182):

“Sm Tregs were also able to show a superior dose-dependent suppressive bystander effect on polyclonal Tconv proliferation compared to mock-Tregs (Extended Data Fig. 1b). This shows that in the presence of SmB/B’₅₈₋₇₂ autoantigen, the Sm Tregs can better suppress Tconvs of other specificities, an effect which contributes to restoring immune tolerance.”

Technical:

The presentation of results as mean \pm SEM is odd. It should be rather SD or quartiles and not SEM.

All graphs which were mean with SEM have been revised to show mean with SD.

In Fig. 11a , mock-transduced Tregs are presented as TCR versus FCS. I believe it should be TCR versus GFP to ensure readers they are GFPnegative.

We thank the reviewer for this point. We have updated the gating in Extended Data Fig 7.

In general, supplementary figures are of bad quality. The size of fonts is too small in many cases or the text is cut (See Fig 11b SSc-H versus ?). The gates used in flow analysis are not clearly displayed.

We have re-made the gating strategies so they are better quality and easier to read. These are now in Extended Data Fig. 6 and 7.4

REVIEWER COMMENTS

Reviewer #1 (Remarks to the Author):

The authors have done a great job in revising the paper. I have no more comments.

Reviewer #2 (Remarks to the Author):

The authors have made a significant effort to improve their manuscript and respond to the reviewer's concerns, which are appreciated and improve the manuscript. Several areas still should be improved upon so that these findings can be better understood within the context of the growing arena of antigen specific Treg generation and therapy. See below comments: The authors note the rarity of autoreactive T cells in the blood of healthy subjects and suggest that it may be more likely those cell isolated using their method would be regulatory. I do not believe that this is supported by many years of data that demonstrated autoreactive T cells in the peripheral blood of HC, using tetramer approaches, as well as AIM based assays, that find effector populations with these specificities in HC. Characterizing TCR1 as one from a regulatory cell based on expression of CD52 is not particularly well founded, or at least not a classically defined nTreg (FOXP3+ CD25hi). The description of CD52++ CD4 T cells (Nature Imm 2013) that is cited specifically cites these cells as distinct from CD4+CD25+Foxp3+ Treg. If the authors would like to support their supposition that this TCR is from a Treg (data in Figure 5) – they should compare their transcriptional profiles to those described for Treg- Pesenacker AM Diabetes 2016 PMID: 26786322 or Pesenacker AM JCI Insight 2019 PMID: 30730852.

The in vitro suppression assays are not convincingly consistent with specificity or bystander suppression. This may be due to the fact that B-LCL are used as APC resulting in a very high level of suppression by polyclonal Mock Treg (figure 8 and supplemental figure1). It would be better to do this with a T cell clone with similar specificity and one with a different specificity to make this point.

The use of patient PBMC to show antigen specific suppression is a good approach, however, it could be much more robust if the authors performed the assay on cells that had been enriched to be responsive to the antigen of interest. An example of this approach can be found in the following paper Yang et al Sci Transl Med. 2022 PMID: 36197963.

Reviewer #3 (Remarks to the Author):

no further comments

REVIEWER COMMENTS

Reviewer #2

Comment 1:

The authors note the rarity of autoreactive T cells in the blood of healthy subjects and suggest that it may be more likely those cell isolated using their method would be regulatory. I do not believe that this is supported by many years of data that demonstrated autoreactive T cells in the peripheral blood of HC, using tetramer approaches, as well as AIM based assays, that find effector populations with these specificities in HC. Characterizing TCR1 as one from a regulatory cell based on expression of CD52 is not particularly well founded, or at least not a classically defined nTreg (FOXP3+ CD25^{hi}). The description of CD52⁺⁺ CD4 T cells (Nature Imm 2013) that is cited specifically cites these cells as distinct from CD4⁺CD25⁺Foxp3⁺ Treg. If the authors would like to support their supposition that this TCR is from a Treg (data in Figure 5) – they should compare their transcriptional profiles to those described for Treg- Pesenacker AM Diabetes 2016 PMID: 26786322 or Pesenacker AM JCI Insight 2019 PMID: 30730852.

We agree with the reviewer's point that CD52 is not a bona fide marker of Tregs and to avoid confusion with Foxp3⁺CD25^{hi} nTregs we have removed references to TCR1⁺ T cells as Tregs in the manuscript, and, instead, referred to the expression of CD52, and the identified upregulated genes, as having potential suppressive capacity:

Previously, Page 5 (lines 99-104)

“Interestingly, the significant upregulation of CD52 in cells expressing TCR1 suggests that TCR1 is associated with a type of nonconventional Treg based on previous research in type 1 diabetes (T1D) showing that CD4⁺CD52^{hi} cells possess Treg properties¹⁴. Other significantly upregulated genes on the TCR1⁺ subset were also associated with Tregs, such as *LAIR2*, expressed on Tregs in other scRNASeq studies, and *IL9R* is associated with Treg suppressive potency^{15,16}.”

Revised version:

“Interestingly, the significant upregulation of CD52 in cells expressing TCR1 suggests that TCR1 is associated with a type of nonconventional **suppressor** T cell based on previous research in type 1 diabetes (T1D) showing that CD4⁺CD52^{hi} cells possess **suppressive** properties¹⁴. Other significantly upregulated genes on the TCR1⁺ subset were also associated with potential **suppressive properties**, such as *LAIR2* and *IL9R*^{15,16}.”

We have also deleted the sentence (page 5, line 104) that states,

“The Treg phenotype of the TCR1 expressing T cell subset reinforces the specificity of TCR1 is for an autoantigen since affinity for self-antigens promotes the selection of Tregs” and removed reference 17 which suggests that TCR derived from Treg cells have increased affinity for self-antigens.

We have also altered our justification for choosing TCR1 to focus on its apparent binding affinity.

Previously (page 6, line 124-127):

Because TCR1 had the greatest apparent affinity for SmB/B'₅₈₋₇₂ bound HLA-DR15 and was associated with CD4⁺CD52^{hi} cells, which have regulatory properties¹⁴, and the upregulation of *LAIR2* and *IL9R*, receptors and molecules associated with Tregs and T cell activation and memory^{22,23}, we selected TCR1 for further testing as a potential Sm-Treg therapy.

Revised sentence:

Because TCR1 had the greatest apparent affinity for SmB/B'₅₈₋₇₂ bound HLA-DR15 and induced T cell activation and memory^{22,23}, we selected TCR1 for further testing as a potential Sm-Treg therapy.

Comment 2:

The in vitro suppression assays are not convincingly consistent with specificity or bystander suppression. This may be due to the fact that B-LCL are used as APC resulting in a very high level of suppression by polyclonal Mock Treg (figure 8 and supplemental figure1). It would be better to do this with a T cell clone with similar specificity and one with a different specificity to make this point. The use of patient PBMC to show antigen specific suppression is a good approach, however, it could be much more robust if the authors performed the assay on cells that had been enriched to be responsive to the antigen of interest. An example of this approach can be found in the following paper Yang et al Sci Transl Med. 2022 PMID: 36197963.

Our in vitro suppression assays are somewhat unconventional and complicated. Admittedly, we have not described them in enough detail for the reviewers (and future readers) to appreciate that the suppression assays were performed on both Sm-specific TCR-transduced Tconvs and untransduced poly-specific Tconvs (bystander Tconvs). Suppression was measured both on the transduced Tconv and the non-transduced poly-specific Tconvs which we refer to as the bystander Tconvs. Meaning that, we have already significantly increased (i.e. enriched) the Tconvs population that are responsive to the antigen of interest and have already measured suppression of Tconv cells with different specificities. To improve the visibility of these methods, we have added more detail to the Results section, Figure Legend 8, Extended Data Figure 7, and added a box within Extended Data Figure 7 to highlight the use of bystander Tconvs.

Previously (page 8, line 168-171):

Based on SmB/B'₅₈₋₇₂ titration, we selected 100 µg/mL SmB/B'₅₈₋₇₂ to pulse HLA-DR15⁺ B-LCLs and co-culture with CD4⁺ T-conventional cells (Tconvs) transduced with Sm-TCR1 and Tregs, either Sm-TCR1-transduced or mock-transduced.

Revised version:

Based on SmB/B'₅₈₋₇₂ titration, we selected 100 µg/mL SmB/B'₅₈₋₇₂ to pulse HLA-DR15⁺ B-LCLs and co-culture with CD4⁺ T-conventional cells (Tconvs) transduced with Sm-TCR1 (to increase the population of Tconvs specific for SmB/B'₅₈₋₇₂) and Tregs, either Sm-TCR1-transduced or mock-transduced.

Previously (page 32 line 745 to 755):

Fig. 8: Sm Tregs are more highly activated and suppress autoreactivity better than polyclonal Tregs.

a, Treg activation measured by CD69 mean fluorescence intensity (MFI) expression by flow cytometry of Tregs transduced with TCR1 (green) or mock transduced (blue) cultured with B-LCLs pulsed with different concentrations of SmB/B'₅₈₋₇₂ in triplicate for 36 hr. **b**, Treg activation of the same assay measured by the Treg-specific activation marker, GARP, expression by flow cytometry. **P*<0.05, ***P*<0.01, ****P*<0.001. **c**, Suppression assay of TCR1-transduced Tconv 5-day proliferation in the presence of SmB/B'₅₈₋₇₂-pulsed DR15⁺ B-LCLs and titrations of either TCR1-transduced Tregs (green) or polyclonal mock Tregs (blue). % Suppression is measured by the increase in CellTrace Violet (CTV) MFI from TCR1-transduced Tconvs from the baseline proliferation with no Tregs. **P*<0.05.

Revised version:

Fig. 8: Sm Tregs are more highly activated and suppress Sm-specific autoreactivity better than polyclonal Tregs.

a, Treg activation measured by CD69 mean fluorescence intensity (MFI) expression by flow cytometry of Tregs transduced with TCR1 (green) or mock transduced (blue) cultured with B-LCLs pulsed with different concentrations of SmB/B'₅₈₋₇₂ in triplicate for 36 hr. **b**, Treg activation of the same assay measured by the Treg-specific activation marker, GARP, expression by flow cytometry. **P*<0.05, ***P*<0.01, ****P*<0.001. **c**, Suppression assay, measured by proliferation, showing the suppressive capacity of Sm-Tregs on Sm-specific Tconv cells transduced with TCR1 in the presence of SmB/B'₅₈₋₇₂-pulsed DR15⁺ B-LCLs and titrations of either TCR1-transduced Tregs (green) or polyclonal mock Tregs (blue). % Suppression is measured by the increase in CellTrace Violet (CTV) MFI from TCR1-transduced Tconvs from the baseline proliferation with no Tregs. **P*<0.05.

Previously (page 35, line 841)

Extended Data Fig. 7: Flow cytometry gating strategy for Sm Treg suppression assay

Suppression assay to assess Treg suppressive capacity of Tconvs was analyzed by gating lymphocytes based on FSC and SSC followed by single cell gating then live cell gating based on Live/dead Near Infra-Red negative cells. T cells (Tregs/Tconvs) were then selected based on CD3⁺ CD19⁻ expression. Tregs were then selected based on CellTrace Far Red (CTFR) positive staining. Tconvs were selected based on CellTrace Violet (CTV) positive staining. The transduced Sm-specific Tconvs were then selected based on GFP⁺ TCR Vbeta21.3⁺ expression and the CTV mean fluorescence intensity of this gate used for suppression analysis.

Revised to:

Extended Data Fig. 7: Flow cytometry gating strategy for Sm Treg suppression assay

Suppression assay to assess Treg suppressive capacity of Tconvs was analyzed by gating lymphocytes based on FSC and SSC followed by single cell gating then live cell gating based on Live/dead Near Infra-Red negative cells. T cells (Tregs/Tconvs) were then selected based on CD3⁺ CD19⁻ expression. Tregs were then selected based on CellTrace Far Red (CTFR) positive staining. Tconvs were selected based on CellTrace Violet (CTV) positive staining. The transduced Sm-specific Tconvs were then selected based on GFP⁺ TCR Vbeta21.3⁺ expression

and the CTV mean fluorescence intensity of this gate used for Sm-specific suppression analysis. The untransduced Sm-specific Tconvs were selected based on GFP- and the CTV mean fluorescence intensity of this gate used for suppression analysis of bystander suppression.

Edited Extended Data Figure 7

REVIEWERS' COMMENTS

Reviewer #2 (Remarks to the Author):

The authors responses to my concerns are adequate as are the changes in the manuscript to reflect that response.